# Global Distribution Prediction of *Cyrtotrachelus buqueti* Guer (Coleoptera: Curculionidae) Insights from the Optimised MaxEnt Model

**DOI:** 10.3390/insects15090708

**Published:** 2024-09-17

**Authors:** Yaqin Peng, Junyi Yang, Danping Xu, Zhihang Zhuo

**Affiliations:** College of Life Science, China West Normal University, Nanchong 637002, China; pengyaqin2023@foxmail.com (Y.P.); yang.junyi@foxmail.com (J.Y.); danpingxu@foxmail.com (D.X.)

**Keywords:** *Cyrtotrachelus buqueti*, MaxEnt, climate change, future climate scenarios, centroid change

## Abstract

**Simple Summary:**

*Cyrtotrachelus buqueti* (*C. buqueti*) primarily feeds on the tender tips of clustered bamboo shoots and is currently one of the major pests in bamboo forests. In this study, the MaxEnt model, a species distribution model, was used to predict the potential current and future global distribution of suitable habitats for *C. buqueti*. The results indicate that climate change significantly impacts the area and geographical distribution of four suitable habitat zones for *C. buqueti*. Under future climate conditions, the area of high, medium, and low suitability zones is expected to increase. In this study, environmental factors affecting the distribution of this species were screened, with temperature and precipitation identified as two key factors likely influencing its distribution. This research provides a scientific basis for preventing the invasion and spread of *C. buqueti*.

**Abstract:**

*Cyrtotrachelus buqueti* Guer is a major pest affecting bamboo forests economically, causing significant damage to bamboo forests in Sichuan Province, China. To understand how *C. buqueti* responds to future climate conditions, an optimized Maximum Entropy Model (Maxent) was used to simulate the potential global distribution patterns of *C. buqueti* under current climate conditions and three different future climate scenarios and to analyze the dominant factors influencing its distribution. The results indicate that Bio18 (precipitation of the warmest quarter), Bio04 (temperature seasonality), Bio06 (minimum temperature of the coldest month), and Bio02 (mean diurnal temperature range) are the main environmental factors affecting the distribution of this species. The global area of high-suitability habitats for *C. buqueti* is 9.00 × 10^4^ km^2^, primarily distributed in China. Under three different future climate scenarios, there are varying degrees of expansion in both the total suitable habitat and the medium-suitability areas for *C. buqueti*. Under the SSP5-8.5 scenario, the medium-suitability area of the species increases the most, reaching 9.83 × 10^4^ km^2^. Additionally, these findings can serve as a reference for developing and implementing control strategies, assisting relevant authorities in more effectively managing and controlling this pest, and mitigating its potential threats to bamboo forest ecosystems and economies.

## 1. Introduction

*Cyrtotrachelus buqueti* Guer is a species of beetle in the genus Cyrtotrachelus within the family Curculionidae. It primarily infests bamboo shoots from 28 genera, including *Bambusa*, *Dendrocalamopsis*, and *Dendrocalamus* [1]. This insect undergoes complete metamorphosis, going through four stages: adult, egg, larva, and pupa. *C. buqueti* lays its eggs in the bamboo shoot sheaths. The larvae feed on the bamboo shoots and pupate in the soil, where they overwinter as adults. In the spring, the adults emerge to feed on bamboo shoots for nutrition. Throughout the year, *C. buqueti* spends 11 months in the soil and approximately 15 days inside bamboo shoots during its one month above ground [2]. Its concealed habits make it difficult to detect and report on population density and the infestation rate of bamboo shoots. *C. buqueti* is an oligophagous insect that primarily feeds on the tender tips of clustered bamboo shoots. Due to its strong concealment, it is hard to detect and challenging to control, making it one of the main pests in bamboo forests. *C. buqueti* is mainly distributed in China’s Sichuan Province, Chongqing Municipality, Guangdong Province, Guangxi Province, and Guizhou Province, as well as in Southeast Asian countries and regions including Vietnam, Myanmar, and Thailand. Wang et al. [3] analyzed the geographical distribution and main occurrence areas of *C. buqueti* in Sichuan Province, China, and found that the southeastern region of Sichuan Province is the most severely affected area by this pest. Recently, Fu et al. [4] predicted the current distribution and future trends of *C. buqueti* in China. The study indicates that Chongqing, Guizhou, and Yunnan are highly suitable distribution areas for this pest. In the future, the distribution center of *C. buqueti* is expected to shift southward [4].

Global warming can alter the distribution of insects worldwide, influencing their habitat range, population density, and migratory behaviors [5]. Based on greenhouse gas emissions, climate warming has become a major trend of climate change in the future, which will directly or indirectly affect ectotherms and insects and their biomes [6]. Climate warming can cause insects to emerge earlier and expand their geographical distribution to higher latitudes and altitudes [7]. Long-term climate warming may even cause genetic mutations in some insect populations [8], ultimately following the natural law of survival of the fittest. Research shows that climate factors have a significant impact on the growth, development, and distribution of *C. buqueti*, with temperature and precipitation being the primary influencing factors [3]. When the maximum temperature reaches 27 °C in September, the population of *C. buqueti* increases rapidly [3]. Additionally, when the daily average temperature exceeds 24 °C, the emergence rate of *C. buqueti* increases [4]. Therefore, temperature directly affects the number of these insects. Currently, *C. buqueti* has become a significant pest affecting clustered bamboo resources in China. In Sichuan Province alone, it infests nearly 67,000 km^2^ annually, with damage rates ranging from 50% to 80% and, in severe cases, reaching up to 100%. It has become a major factor limiting the development of bamboo forests for paper production [9]. Therefore, to monitor the global spread of *C. buqueti* in a timely manner, it is important to establish an effective prevention and monitoring system and to study the potentially suitable habitats for *C. buqueti* worldwide.

Species distribution models (SDMs) are important tools in ecology and biogeography research, commonly used to relate specific species to ecological niche factors [10]. These models help scientists understand and predict species distribution under various environmental conditions, and they are crucial for biodiversity conservation and ecosystem management. Today, many SDMs have been developed, such as MaxEnt [11], GARP [12], BIOCLIM [13], and Climex [14], to predict species’ geographical distributions. Among these, the MaxEnt model has become one of the most popular species distribution and ecological niche models due to its efficiency and accuracy [15].

The MaxEnt model, based on the principle of maximum entropy, can predict the potential distribution of a species in other areas with similar climatic conditions, even with limited species distribution data [15]. It has the following notable advantages: (i) high success rate, with statistical significance even with small sample sizes [11]; (ii) capability to handle both continuous and categorical environmental variables [16]; and (iii) no absence points, which can be costly/difficult to get thorough datasets on [11,16]. Currently, MaxEnt is widely used to predict suitable areas for various biological and non-biological entities. Specific applications include predicting the distribution of endangered species [17] to provide a scientific basis for conservation measures, forecasting the potential distribution of invasive species [18] to help develop prevention and control strategies, and identifying soil pollution areas to assist in environmental management [19]. Additionally, MaxEnt is used to study the impact of climate change on species distribution by simulating potential distributions under different climate scenarios to assess the risks faced by species. The MaxEnt model plays an increasingly important role in biogeography and ecology research, becoming a crucial tool for predicting and managing species distribution.

Based on the distribution data of *C. buqueti*, the potential global distribution, spatial patterns, and suitable habitats of *C. buqueti* under current and future climate scenarios were predicted using ArcGIS tools and the MaxEnt model. The research objectives primarily focus on the following aspects: (1). Potential global distribution of *C. buqueti* under current and future climate scenarios: using the MaxEnt model, determine the geographical regions where *C. buqueti* may occur under different climate conditions, providing a scientific basis for monitoring the species. (2). Impact of environmental factors on the suitable habitat distribution of *C. buqueti*: analyze the effects of various environmental factors (such as temperature, precipitation, elevation, etc.) on *C. buqueti*’s distribution to identify the most critical environmental factors influencing its habitat selection. This part of the research will help predict the long-term impacts of climate change on *C. buqueti* and its ecosystem, providing references for developing response strategies.

## 2. Materials and Methods

### 2.1. Source of Species Data

Data related to *C. buqueti* are sourced from the Global Biodiversity Information Facility (GBIF) (http://www.gbif.org, accessed date: 31 July 2024) and the relevant literature. After screening, a total of 379 species records were obtained, including those with latitude and longitude coordinates from GBIF [20] and data from the relevant literature [3] (Appendix A). Although MaxEnt can still demonstrate good predictive ability with small sample sizes, studies have shown that small sample sizes may lead to greater prediction errors, and repeated or similar occurrence data may result in overfitting [21]. Therefore, only the filtered occurrence data can be used for model prediction. To ensure data independence and reduce spatial autocorrelation of sampling points, we used ENMTools v1.4 [22] (https://github.com/danlwarren/ENMTools, version 1.4, accessed on 27 August 2024) to filter the distribution points. At a spatial resolution of 2.5 arcminutes (approximately 4.5 km), only one occurrence point was retained per grid cell [23]. Ultimately, 374 occurrence records of *C. buqueti* were retained for analysis (Figure 1).

### 2.2. Environmental Variables Related to C. buqueti

In this study, the data for 19 bioclimatic variables (2.5 arcminutes) for the current period (1970–2000) and future decades (2040s: 2041–2060, and 2070s: 2061–2080) were sourced from the Worldclim-Global Climate Database (WorldClim, http://worldclim.org/, accessed date: 12 November 2023, Version 2.1) [24]. Using the SDM tools (http://www.sdmtoolbox.org/technical-info, Version 10.4 to 10.9, accessed date: 12 November 2023) in ArcGIS v10.8 (ESRI Inc., Redlands, CA, USA), these bioclimatic data were converted from TIF file format to ASCII file format [25]. The topographic data were sourced from the National Centers for Environmental Information (NOAA NCEI, https://www.ngdc.noaa.gov/, accessed date: 12 November 2023), including three global elevation (DEM) datasets, all with a resolution of 2.5 arcminutes to ensure clear distribution maps. In this study, the BCC-CSM2-MR model from the Sixth Coupled Model Intercomparison Project (CMIP6) was used, considering three Shared Socioeconomic Pathways (SSPs) scenarios: a low carbon emission scenario (SSP1-2.6), a medium to high carbon emission scenario (SSP3-7.0), and a high carbon emission scenario (SSP5-8.5) [26,27].

Nineteen bioclimatic and three terrain variables (Table 1) were considered when building the models for *C. buqueti*. Since not all environmental variables impact species distribution, selecting relevant environmental factors specific to the species and removing those that contribute little to the prediction of species distribution can enhance the model’s accuracy [4]. This allows the model to more accurately reflect the species’ actual ecological requirements and living conditions. Additionally, selecting relevant environmental factors can reduce the model’s complexity and computational burden, making the model more streamlined and efficient. This approach allows for a more accurate reflection of the species’ actual ecological requirements and living conditions. Additionally, eliminating irrelevant environmental factors can reduce the model’s complexity and computational burden, making the model more streamlined and efficient. First, 394 distribution points of the species and all environmental variables were imported into the MaxEnt software (https://biodiversityinformatics.amnh.org/open_source/maxent/, version 3.4.1, accessed date: 1 July 2023). The contribution rates of the environmental variables were calculated, and variables with contribution rates lower than 1% were removed. Next, Pearson correlation coefficient analysis was conducted to assess the correlations between environmental factors. Environmental factors with Pearson coefficients greater than |0.8| were removed due to high correlation [28]. Ultimately, out of 22 environmental variables, nine were selected to predict the suitable habitat of *C. buqueti* (Table 2).

### 2.3. Optimization of Model Parameters

In this study, the MaxEnt model was used to predict the potential distribution of *C. buqueti* for the current period, the 2050s and the 2070s. The predictive performance of the model is primarily influenced by two parameters: the feature class (FC) and the regularization multiplier (RM). In MaxEnt software, there are five feature parameters: linear (L), quadratic (Q), product (P), threshold (T), and hinge (H). The RM value directly affects the concentration of the model’s predictions; a smaller RM value may result in more restricted and concentrated predictions, while a larger RM value may lead to broader and more dispersed predictions [11,23]. In this study, when adjusting the optimal parameters, the range for the RM value was set from 0.1 to 4, with a step size of 0.1 [29].

The R package ENMeval, developed by Muscarella et al. [30], was used to optimize the model parameters. This package selects the most suitable model parameters by evaluating different combinations of feature class and regularization multiplier parameters. Specifically, in the R package “kuenm” [30], the regularization multiplier parameters were combined with feature class combinations to create 1240 candidate models in the software. Parameter combinations with significance, omission rates below 5%, and delta AICc (the difference in the Akaike Information Criterion corrected for small sample sizes between the calibrated optimal model and the current model) less than or equal to two were used to run the MaxEnt model. This selection criteria ensures that the chosen model parameter combinations not only significantly enhance the model’s predictive ability but also maintain a low omission rate, thus ensuring the model’s robustness [31,32]. At the same time, the delta AICc ≤ 2 criterion helps to select models with performance close to the optimal model, thereby avoiding excessive model complexity [32]. Through this systematic parameter optimization process, the optimal model settings were determined, enhancing the accuracy and reliability of species distribution predictions. Ultimately, MaxEnt software was used with a feature class combination of t and a regularization multiplier of 0.4 to establish the MaxEnt model.

### 2.4. Maxent Model Construction and Validation

The 374 distribution point data and nine environmental variables were imported into MaxEnt software v3.4.1 for modeling analysis. In this study, 25% of the distribution data were randomly selected as test data, while the remaining 75% were used as training data. The maximum number of background points is 10,000. The repeat type is a subsample. The performance of the MaxEnt model is typically evaluated using the AUC (Area Under the Receiver Operating Characteristic Curve). An AUC value of 0.5–0.7 indicates poor performance, 0.7–0.9 indicates moderate performance, and greater than 0.9 indicates high performance [27].

### 2.5. Statistical Analysis and Suitable Habitat Grade Classification

To improve model performance, the evaluation process was repeated with 10-fold cross-validation. The model predictions were then imported into a Geographic Information System (GIS) and classified using the reclassification tool in ArcMap 10.8. Based on the literature and the habitat adaptability of the species [3,33,34], four potential habitat types are classified as follows: high habitat suitability (0.5 ≤ *p* ≤ 1), medium habitat suitability (0.3 ≤ *p* < 0.5), low habitat suitability (0.05 ≤ *p* < 0.3); unsuitable habitat (*p* < 0.05).

## 3. Result

### 3.1. Model Optimization Results Validation and Variable Selection

Based on the ROC curve generated by the Maxent model and the calculated AUC value, the AUC value obtained from the averaged model is 0.983 (Figure 2). This result indicates that the model has good predictive performance and can effectively predict the suitable habitat for *C. buqueti*. Under current climatic conditions, the nine variables involved in constructing the MaxEnt model for *C. buqueti* are Bio18, Bio04, Bio02, Bio15, Bio19, Bio17, Bio06, Elev, and Aspect. Among these nine variables, Bio18 (66.4%), Bio04 (14.4%), and Bio02 (9.1%) contribute the most to the model (Table 3).

The environmental conditions when the species’ habitat adaptability threshold exceeds 0.5 align with the environmental conditions of the species’ highly suitable habitat. Based on the jackknife test (Figure 3), Bio18, Bio04, and Bio06 have a significant impact on the suitability of species distribution. The threshold values of the response curves indicate that under these environmental conditions, *C. buqueti* is most suited for survival and reproduction. Considering the contribution rates of the environmental factors and the jackknife plots, the results show that when the environmental conditions are as follows: precipitation of the warmest quarter (Bio18) between 656.43–959.68 mm, temperature seasonality (standard deviation × 100) (Bio04) between 687.11–705.94, minimum temperature of the coldest month (Bio06) between 1.95 °C and 4.32 °C, and mean diurnal temperature range (Bio02) between 6.078 °C and 7.370 °C, the conditions are optimal for the survival of *C. buqueti* (Figure 4).

### 3.2. Prediction of Potential Geographic Distribution of C. buqueti under Current Climatic Conditions

The Maxent model was used to predict the global suitable habitat distribution of *C. buqueti* and then visualize the results using ArcGIS v10.8. *C. buqueti* has a limited global distribution (Figure 5), with low-suitability areas mainly concentrated in eight countries: a small part of southern Bhutan, China, a small area in northeastern India, southern Japan, a small part of northwestern Myanmar, southern North Korea, southern South Korea, and northern Vietnam. The total area of low-suitability regions is 19.81 × 10^4^ km^2^. Both the medium-suitability and high-suitability areas are located in China, covering 4.61 × 10^4^ km^2^ and 9.00 × 10^4^ km^2^, respectively. The high-suitability areas are primarily distributed in the southeastern part of Sichuan Province, northwestern Chongqing, central Guizhou Province, and the northeastern part of Yunnan Province bordering Sichuan. The medium-suitability areas are mainly distributed in a band surrounding the high-suitability areas (Table 4).

### 3.3. Potential Habitat Changes of C. buqueti under Future Climate Scenarios

In this study, not only was the current suitable habitat distribution of *C. buqueti* predicted, but its suitable habitat distribution for the 2050s and 2070s was also forecast (Figure 6). In the future, the medium-suitability and high-suitability areas for *C. buqueti* will primarily be distributed in China. The area of medium-suitability regions changes differently under different scenarios, while the area of high-suitability regions will decrease to varying extents. Under the SSP1-2.6 scenario, the medium-suitability area for *C. buqueti* initially decreases to 3.01 × 10^4^ km^2^, then increases to 8.41 × 10^4^ km^2^. Under the SSP5-8.5 scenario, this medium-suitability area increases the most, reaching 9.83 × 10^4^ km^2^. Under the SSP1-2.6 scenario, the high-suitability area for *C. buqueti* decreases first and then increases, reducing to 3.95 × 10^4^ km^2^ in the 2050s and expanding to 7.55 × 10^4^ km^2^ in the 2070s (Table 4). Under the SSP3-7.0 and SSP5-8.5 scenarios, the high-suitability area for this species is smaller than the current high-suitability area, with only minor changes.

The suitable habitat for this species has a limited global distribution, primarily concentrated in China. According to Figure 7, its range undergoes various changes under future climate scenarios. Under future climate conditions, the main distribution areas of *C. buqueti* remain close to their current locations. In the SSP1-2.6 scenario, the medium-suitability and high-suitability areas for *C. buqueti* expand eastward. In the SSP3-7.0 scenario, the medium-suitability area in eastern Sichuan Province decreases significantly, while the high-suitability area in Guizhou Province expands slightly to the east. In the SSP5-8.5 scenario, from the 2050s to the 2070s, both the medium-suitability and high-suitability areas for *C. buqueti* expand northward.

### 3.4. Centroid Changes in Potential Distribution

In this study, the predicted centroid shifts of *C. buqueti*’s high-suitability areas vary under different scenarios (Figure 8). Under the current scenario, the distribution center of *C. buqueti*’s high-suitability area is located in Dongxing District, Neijiang City, Sichuan Province, China (29°28′56″ E, 105°1′48″ N). In the future, the centroid shift direction of this species’ high-suitability area remains similar across scenarios. Under the SSP1-2.6 scenario, the centroid first shifts southwestward and then southeastward, with a relatively small shift, reaching Xingwen County, Yibin City, Sichuan Province (28°15′11″ E, 105°11′20″ N). Under the SSP3-7.0 and SSP5-8.5 scenarios, the centroid shifts southeastward to Guizhou Province, with coordinates 27°39′58″ E, 106°12′4″ N and 27°34′28″ E, 105°41′48″ N, respectively. The movement trajectories under these two emission scenarios are similar, with the greatest southeastward shift occurring under the SSP3-7.0 scenario by the 2070s.

## 4. Discussions

Under the three Shared Socioeconomic Pathways (SSPs) for the future, the environmental factors used in the model include a low-carbon emission scenario (SSP1-2.6), a medium-to-high carbon emission scenario (SSP3-7.0), and a high-carbon emission scenario (SSP5-8.5). These scenarios represent projections of future carbon emissions due to climate change rather than precise evaluations of future emissions. When researchers use the MaxEnt model to predict species distribution, they have two options: using default model parameters or optimized model parameters. The default model parameters in MaxEnt were developed by early researchers based on extensive experimental data to simplify the MaxEnt model and provide a set of parameters that generally work well across a range of conditions [35]. However, these default parameters often lack transferability and may not be suitable for predicting the future potential distribution of species [16,36]. Therefore, after comparing significance and data omission rates, the MaxEnt model parameters used in this study were optimized and found to be appropriate for predicting the distribution of *C. buqueti*.

Combining the contribution rates of the MaxEnt model with the Jackknife test, it is concluded that the precipitation of the warmest quarter, temperature seasonality (standard deviation × 100), minimum temperature of the coldest month, and mean diurnal temperature range are important environmental factors influencing the distribution of *C. buqueti*. In summary, temperature and precipitation can affect the distribution of *C. buqueti*. The precipitation of the warmest quarter has the highest contribution rate, indicating that precipitation related to climate change primarily affects the distribution of *C. buqueti*. In China, adult *C. buqueti* emerge from the soil between June and August and feed on bamboo shoots. June to August corresponds to summer in the Northern Hemisphere, and excessive rainfall during this period can lead to high soil moisture and root rot in bamboo, which is unfavorable for the survival and foraging of this pest. Conversely, when precipitation is too low, the soil becomes compacted, hindering bamboo growth and affecting the survival of this species. Studies have shown that the suitable summer precipitation range for bamboo forests in China is 496–705 mm [37], which aligns with the range suitable for the survival of *C. buqueti*. *C. buqueti* prefers warm and humid environments and overwinters in pupal chambers in the soil during winter. Although the soil provides some insulation, excessively low temperatures can still affect the insect’s body temperature, posing a threat to its survival. This corresponds with the finding that *C. buqueti* thrives in conditions where the minimum temperature of the coldest month (Bio06) is between 1.95 °C and 4.32 °C. As ectothermic animals, insects are directly affected by significant daily temperature fluctuations, which can severely impact their survival.

Based on the MaxEnt model, the current and future global distribution of suitable habitats for *C. buqueti* has been predicted. The low habitat suitability is primarily distributed across eight countries, while the medium and high habitat suitability are mainly located in China. Through searches on GBIF and in the literature, *C. buqueti* has been found to primarily inhabit China and Thailand in Asia (as shown in the Figure 1). The distribution points of this species closely align with the predicted suitable areas, indicating that the predictions are fairly accurate. Based on these predictions, *C. buqueti* likely originated in China and has not yet widely invaded other countries. Bamboo resources are mainly concentrated in tropical and subtropical low-latitude regions, and *C. buqueti*, being a stenophagous pest, has limited migration and dispersal abilities. Consequently, its current global distribution is relatively narrow. The high-suitability areas for *C. buqueti* are primarily concentrated in eastern Sichuan Province, northwestern Chongqing, and central Guizhou Province in China. These three regions are mainly influenced by a subtropical monsoon climate, characterized by hot, rainy summers and mild, dry winters, which are conducive to the survival of this species.

To prevent the spread of *C. buqueti*, relevant authorities need to adopt a comprehensive set of strategies, including preventive measures, physical control, chemical control, biological control, and ecological management, in order to form an integrated control system. Based on the characteristics of *C. buqueti*, the following control recommendations are proposed: (1) Prevention-focused approach: On one hand, efforts should be made to prevent the spread to currently uninfected areas, while on the other hand, potential future high-suitability areas for *C. buqueti* should also be protected. It is important to maintain the health of bamboo forests by regularly removing diseased or weakened bamboo, ensuring good ventilation in the forest, and reducing over-planting. *C. buqueti* is an oligophagous pest that feeds only on bamboo. Currently, the eastern part of Sichuan Province and central Guizhou are high-suitability areas for *C. buqueti*, where bamboo forests may already be infested. If bamboo is to be transplanted from these areas, strict disinfection and pest control measures must be applied. Additionally, the core of *C. buqueti*’s high-suitability area is predicted to shift southeastward in the future. Therefore, central Guizhou and southwest Chongqing may become high-risk areas for *C. buqueti* in the future, and special attention should be given to strengthening control measures in these regions. (2) Timely control in infested areas: In already infested regions, control measures should be promptly implemented, such as manual removal, trap-based capture, insecticide spraying, and tree trunk injections of insecticides. Based on the predicted distribution of *C. buqueti*’s suitable habitat from this study, self-inspections can be carried out in these areas. For example, data on *C. buqueti*’s occurrence show that this species is primarily distributed in the eastern part of Sichuan Province. However, according to the predictions from this study, parts of the Sichuan-Chongqing border area and central Guizhou are also high-suitability zones for this pest. Therefore, it is necessary to monitor these areas, and if *C. buqueti* is detected, physical, chemical, or biological control measures should be promptly employed.

## 5. Conclusions

Based on 374 occurrence data points for *C. buqueti* and nine major environmental factors, the global occurrence patterns of *C. buqueti* have been summarized. Given the data points and predictions from this study, the potential distribution of the species’ suitable habitats has been effectively identified. The results indicate that the species’ suitable habitats are currently mainly distributed in southeastern Sichuan Province, northwestern Chongqing, central Guizhou Province, and the northeastern part of Yunnan Province bordering Sichuan. Under future climate warming conditions, there is a trend for these highly suitable areas to expand and shift southeastward. To monitor the distribution of *C. buqueti* early on, it is advisable to combine the predicted suitable habitats from this study with field monitoring to check for the presence of this species in these areas and to implement preventive measures in suitable areas where its presence has not yet been detected.

## Figures and Tables

**Figure 1 insects-15-00708-f001:**
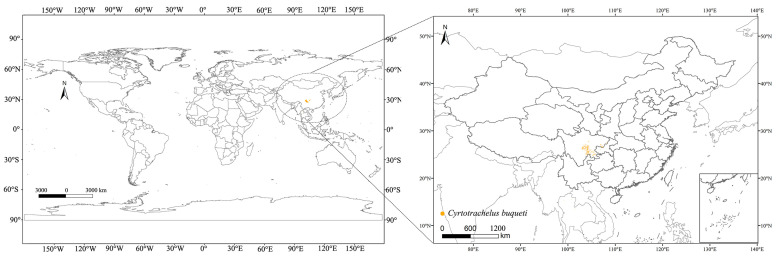
Geographical distribution points of *C. buqueti* in the world.

**Figure 2 insects-15-00708-f002:**
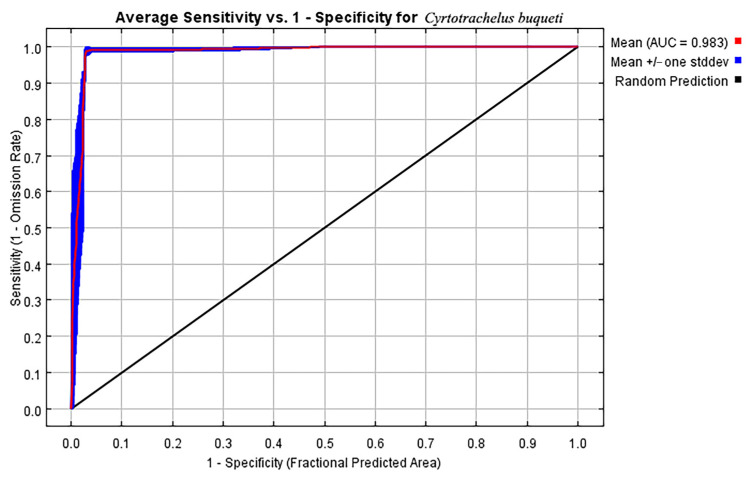
The receiver operating characteristic (ROC) curve generated by the MaxEnt model.

**Figure 3 insects-15-00708-f003:**
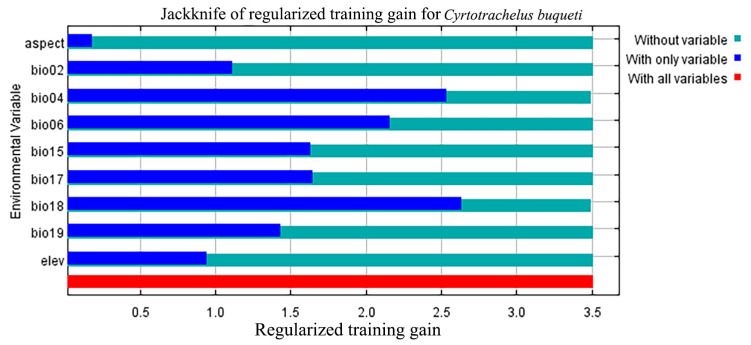
Jackknife test of variable importance for the MaxEnt model of *C. buqueti* distribution.

**Figure 4 insects-15-00708-f004:**
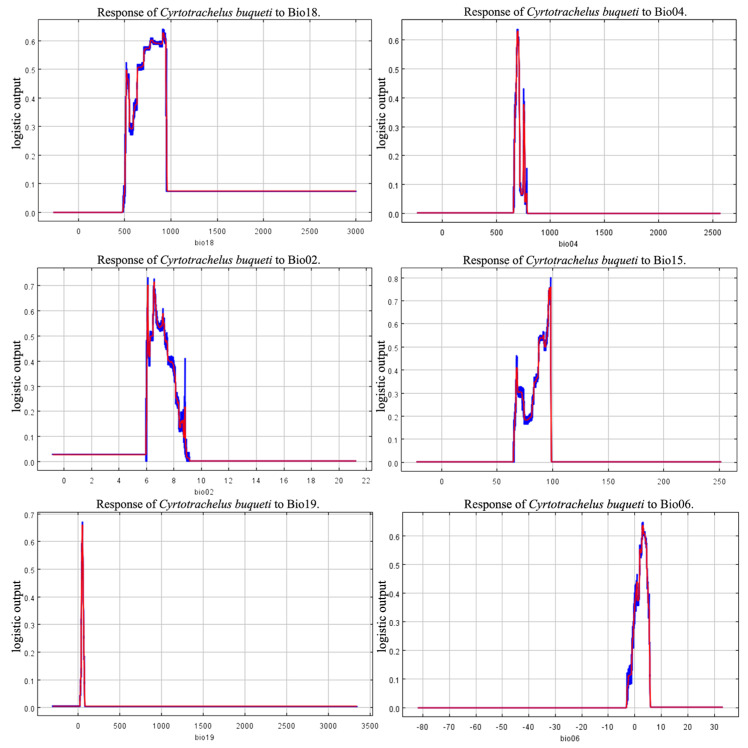
Response curves of five environmental variables.

**Figure 5 insects-15-00708-f005:**
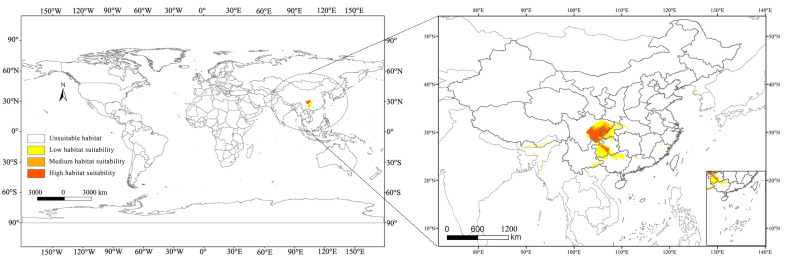
Distribution of suitable habitats for *C. buqueti* under the current climate.

**Figure 6 insects-15-00708-f006:**
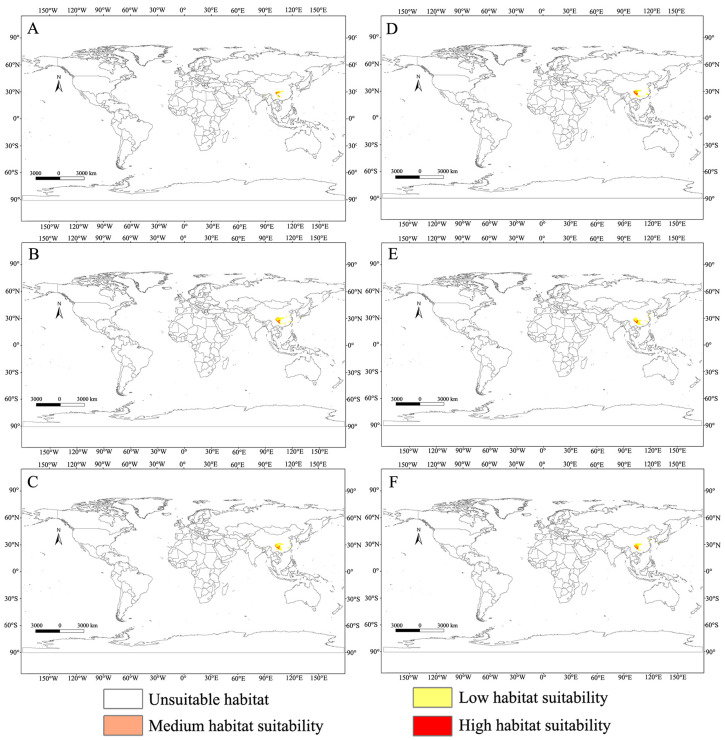
Global potential distribution map for *C. buqueti* under the future climate scenario. (**A**) SSP1-2.6 in the year 2050. (**B**) SSP3-7.0 in the year 2050. (**C**) SSP5-8.5 in the year 2050. (**D**) SSP1-2.6 in the year 2070. (**E**) SSP3-7.0 in the year 2070. (**F**) SSP5-8.5 in the year 2070.

**Figure 7 insects-15-00708-f007:**
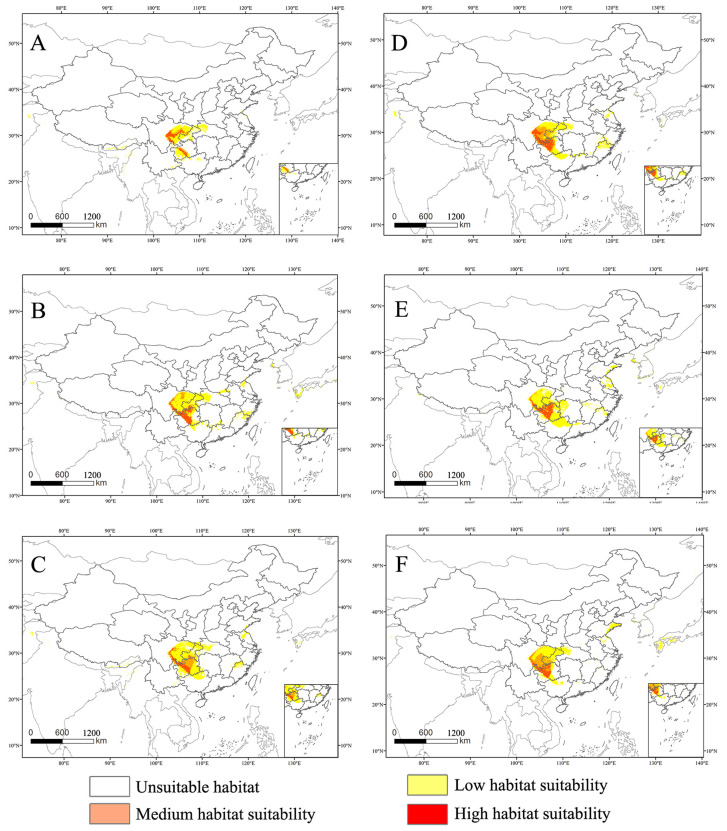
The high habitat suitability for *C. buqueti* under the future climate scenario. (**A**) SSP1-2.6 in the year 2050. (**B**) SSP3-7.0 in the year 2050. (**C**) SSP5-8.5 in the year 2050. (**D**) SSP1-2.6 in the year 2070. (**E**) SSP3-7.0 in the year 2070. (**F**) SSP5-8.5 in the year 2070.

**Figure 8 insects-15-00708-f008:**
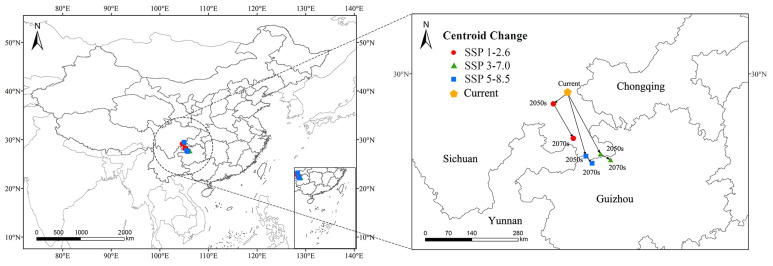
Changes in the centroid of the potential highly suitable distribution of *C. buqueti* worldwide.

**Table 1 insects-15-00708-t001:** Environmental variables related to the distributions.

Abbreviation	Climate Variables	Unit
Bio01	Annual mean temperature	°C
Bio02	Mean diurnal temperature range	°C
Bio03	Isothermality (bio2/bio7) (×100)	
Bio04	Temperature Seasonalit (standard deviation × 100)	
Bio05	Max temperature of warmest month	°C
Bio06	Min temperature of coldest month	°C
Bio07	Temperature annual range (bio5–bio6)	°C
Bio08	Mean temperature of wettest quarter	°C
Bio09	Mean temperature of driest quarter	°C
Bio10	Mean temperature of warmest quarter	°C
Bio11	Mean temperature of coldest quarter	°C
Bio12	Annual precipitation	mm
Bio13	Precipitation of wettest month	mm
Bio14	Precipitation of driest month	mm
Bio15	Precipitation seasonality (Coefficient of variation)	
Bio16	Precipitation of wettest quarter	mm
Bio17	Precipitation of driest quarter	mm
Bio18	Precipitation of warmest quarter	mm
Bio19	Precipitation of coldest quarter	mm
Elev	Altitude (elevation above sea level) (m)	m
Slope	Slope	°
Aspect	Aspect	rad

**Table 2 insects-15-00708-t002:** The nine environment variables used for modeling.

Abbreviation	Climate Variables	Unit
Bio02	Mean diurnal range	°C
Bio04	Temperature Seasonalit (standard deviation × 100)	
Bio06	Min temperature of coldest month	°C
Bio15	Precipitation seasonality (Coefficient of variation)	
Bio17	Precipitation of driest quarter	mm
Bio18	Precipitation of warmest quarter	mm
Bio19	Precipitation of coldest quarter	mm
Elev	Altitude (elevation above sea level) (m)	m
Aspect	Aspect	rad

**Table 3 insects-15-00708-t003:** Contribution and permutation are important in the estimation of climate variables in the MaxEnt model of *Cyrtotrachelus buqueti*.

Variable	Percent Contribution	Permutation Importance
Bio18	66.4	75.4
Bio04	14.4	8.7
Bio02	9.1	2
Bio15	5.3	2.5
Bio19	3.6	7.7
Bio17	0.7	2.6
Bio06	0.6	0.5
Elev	0.1	0.7
Aspect	0	0

**Table 4 insects-15-00708-t004:** Prediction of suitable areas for *Cyrtotrachelus buqueti* under current and future climatic conditions.

Scenarios	Decade	Total Suitable Regions		Regions of Low Habitat Suitability		Regions of Medium Habitat Suitability		Regions of High Habitat Suitability	
		Area (10^4^ km^2^)	Area Change (%)	Area (10^4^ km^2^)	Area Change (%)	Area (10^4^ km^2^)	Area Change (%)	Area (10^4^ km^2^)	Area Change (%)
-	Current	33.42	-	19.81	-	4.61	-	9.00	-
SSP1-2.6	2050s	26.00	−22.20	19.03	−3.94	3.01	−34.71	3.95	−56.11
	2070s	41.10	22.98	25.15	26.96	8.41	82.43	7.55	−16.11
SSP3-7.0	2050s	44.52	33.21	34.26	72.94	3.51	−23.86	6.75	−25.00
	2070s	45.86	37.22	37.36	88.59	1.83	−60.30	6.67	−25.89
SSP5-8.5	2050s	46.11	37.97	35.23	77.84	6.34	37.53	4.54	−49.56
	2070s	41.00	22.68	26.12	31.85	9.83	113.23	5.04	−44.00

## Data Availability

The data supporting the results are available in a public repository at GBIF.org (31 July 2024) GBIF Occurrence Download https://doi.org/10.15468/dl.msv3qz, accessed on 31 July 2024 and Peng, Yaqin (2024). *Cyrtotrachelus buquet* distributions. figshare. Dataset. https://doi.org/10.6084/m9.figshare.26412433.v2, accessed on 31 July 2024.

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
