# Peer review of "Global Distribution Prediction of Cyrtotrachelus buqueti Guer (Coleoptera: Curculionidae) Insights from the Optimised MaxEnt Model"

_insects, 2024, doi:10.3390/insects15090708_

Round 1
Reviewer 1 Report
Comments and Suggestions for Authors
The manuscript is poorly written, and there are flaws in the research methods. There are two key issues that need to be revised, and the manuscript cannot be accepted without these issues being resolved. The specific issues are as follows:
1. Lines 163-179 in Materials and Methods: The reason for selecting environmental variables is mainly because the correlation between environmental variables can affect the accuracy of the model, and these methods cannot determine which variables are unrelated to species distribution, so I doubt this method.
And only 2 out of 19 bioclimatic variables were retained, while all 3 topographic variables were retained. The prediction results also showed that the influence of bioclimatic variables was greater than that of topographic factors, and some important bioclimatic variables were likely excluded.
In addition, the existence of highly correlated environmental variables is not about excluding them, but about selecting one that is more important.
I think it is necessary to change the method of screening environmental variables and re model. I am skeptical about the accuracy of the predicted results.
2. Lines 232-238 in Materials and Methods: Why choose 0.05 as the threshold for suitable habitat? This seems unreasonable. This method refers to research literature on Agastache rugosa, and I believe there are significant differences between this species and the species studied in this study. Why should this study be referenced?
Did the model in this study run 10 times? Which one was chosen as the final result? According to the research results, it seems that the average value was chosen as the final result, and it seems that the result with the highest AUC should be selected as the final result.
The introduction and discussion are not well written, and there are still many minor issues. The author seems to have not taken this manuscript seriously. Many citations in this study are inappropriate, and the author seems to not know how to cite references. The specific issues are as follows:
Page 1
3. Title: The “Cyrotrachelus buqueti Guer” should be changed to “Cyrotrachelus buqueti Guer (Coleoptera: Curculionidae)”.
4. Line 7 in page 1: The word is repeated.
5. Line 10 in page 1: This content is repetitive with the content on line 6.
6. Lines 20-21 in page 1: It needs to be explained which period the result is, I think it is the result of the current period.
Introduction
7. Lines 31-32: The “Cyrotrachelus buqueti” and “Cyrotrachelus” need to be italicized.
8. Lines 47-50, 50-53, 53-56, 107-109, 109-112: The citation should only include the surname, and the citation should be after the person's name rather than at the end of the sentence. For example, “Jiawen Wang et al. analyzed the geographical distribution and main occurrence areas of C. buqueti in Sichuan Province, China, and found that the southeastern region of Sichuan Province is the most severely affected area by this pest[3].” should be changed to “Wang et al. [3] analyzed the geographical distribution and main occurrence areas of C. buqueti in Sichuan Province, China, and found that the southeastern region of Sichuan Province is the most severely affected area by this pest.”.
9. Lines 59-61, 61-63, 65-67, 67-68, 68-69, 84-86, 93-94, 94-98, 98-100, 100-103, 103-105, 207: These contents require citations.
10. Lines 77-118: This paragraph is too long and needs to be segmented, and there is too much introduction about the MaxEnt model, which needs to be streamlined.
11. Lines 82-84, 87-90: These conclusions require citations, and this content overlaps with the content on lines 90-93.
12. Lines 86-87: This sentence seems unrelated to the content of this paragraph, it is recommended to delete it.
13. Lines 90-93: Each point of these contents needs to be accompanied by a citation, not just at the end of the sentence.
14. Lines 107-112: Why are these references cited? They seem useless and it is recommended to delete them.
15. Lines 113-114: What are the challenges? Is that the content of the next sentence? These are not clearly stated and lack citations.
16. Lines 122-132: The format and capitalization of these contents need to be checked, and many of them are incorrect.
17. Line 127: This study does not seem to include soil type data, but rather terrain data.
Materials and Methods
18. Line 136: What literature did you obtain the distribution data of C. buqueti from? These literature need to be cited. Did you obtain distribution data of C. buqueti from this literature? This literature seems unrelated to C. buqueti. According to Data availability, it seems that only distribution data from GBIF has been obtained, and the data source should be explained truthfully, not made up.
19. Lines 136, 150, 153: The “website:” should be deleted. These links in the manuscript need to add access times, which can be referenced to the Data availability.
20. Lines 143, 151, 171, 173, 207, 210, 222: These software need to provide citations or links or information about the company, country, etc. they belong to. For example, “ArcGIS 10.8” can be changed to “ArcGIS v10.8 (ESRI Inc., CA, USA)”. Additionally, these software require corresponding version numbers to be provided.
21. Lines 145-146: The occurrence records of C. buqueti need to be provided and indicate which ones come from GBIF and which ones come from literature.
22. Line 148: The resolution of bioclimatic data needs to be stated.
23. Lines 178-179, 247-248, 288, 301: The format is incorrect, and the “(Table 2)” should be changed to “(Table 2)”.
23. Lines 181-196: These seem to need to be explained in the introduction, and only a brief introduction is needed in the materials and methods. It is suggested to delete or condense them into 1-2 sentences.
24. Lines 208-210: This citation is incorrect, and it is not the article that developed this software package.
25. Line 224: The title is not accurate, as this section includes the establishment and evaluation of the model.
26. Line 226: The link is incorrect, the software cannot be found.
Result in pages 6-11
27. Lines 244-246: This sentence seems redundant, it is recommended to delete it.
28. Lines 250-259: The format of the table have issues and need to be modified.
29. Lines 273-274: The Latin name in the figure is written incorrectly, 'Guer' does not need to be italicized
30. Lines 275-288: The format of the title needs to be modified. The expression of the content is seriously incorrect. The result of this paragraph is the location of the suitable area, not the distribution location of C. buqueti. These regions are only areas where C. buqueti may be distributed, but they may not necessarily have its distribution.
31. Lines 291-301: The description of this section is insufficient, and it is necessary to clarify the approximate location of the suitable area for each period, as well as the differences from the current period, rather than simply discussing changes in the area of suitable areas.
32. Lines 303-305: The annotation of the figure needs clarification, as there is no 'A', 'B', 'C', 'D', 'E', or 'F' in the figure. The figure is small and not easy to observe.
Result in page 12
33. From here on, the line numbers are discontinuous from the previous text.
Discussions
34. Line 19: This sentence is a bit confusing, I think maybe “The predictions from the MaxEnt model vary under different climate scenarios.” should be changed to “The prediction results of MaxEnt model vary under different climate scenarios.”.
35. Lines 25-26: This sentence seems unrelated to the content of this paragraph, it is recommended to delete it.
36. Lines 27-29: There is only one citation here, which does not seem to support the conclusion that the model is widely used.
37. Line 43: The "the average temperature during the wettest quarter" should use the standard name "mean temperature of wettest quarter", which can be found in: https://www.worldclim.org/data/bioclim.html. These variables have standard names and cannot be changed. Other areas that need to be modified need to be modified by the author themselves and will not be indicated.
38. Lines 43-44: The citation should only include the surname, and the citation should be after the person's name rather than at the end of the sentence.
39. Lines 51-64: These contents simply describe the research results, which have already been explained within the results and should not be repeated. The reasons for these phenomena should be explained, but the author did not explain them at all.
40. Lines 53-54: This study does not support this conclusion.
41. Lines 65-98: This paragraph is too long and needs to be segmented. This paragraph needs to be streamlined. Although the control of this species is important, these methods are irrelevant to the conclusions of this study. These contents can explain, but they should not occupy such a large space. Specific prevention and control plans should be proposed based on this study.
Conclusions
42. Lines 102-105, 112-115: These visions are great, but has this study solved these problems? The author needs to carefully consider and revise the manuscript to achieve these goals.
Comments on the Quality of English LanguageThere doesn't seem to be a major issue with the language, but there are many issues with the wording, which have been explained in the comments.
Author Response
Comments 1: Lines 163-179 in Materials and Methods: The reason for selecting environmental variables is mainly because the correlation between environmental variables can affect the accuracy of the model, and these methods cannot determine which variables are unrelated to species distribution, so I doubt this method. And only 2 out of 19 bioclimatic variables were retained, while all 3 topographic variables were retained. The prediction results also showed that the influence of bioclimatic variables was greater than that of topographic factors, and some important bioclimatic variables were likely excluded.
In addition, the existence of highly correlated environmental variables is not about excluding them, but about selecting one that is more important. I think it is necessary to change the method of screening environmental variables and re model. I am skeptical about the accuracy of the predicted results. |
Response 1: Thank you for pointing this out. In this study, VIF (Variance Inflation Factor), Pearson correlation coefficient, and contribution rate were used to screen environmental factors. VIF indicates the degree of multicollinearity (correlation between predictor variables) in regression analysis. When calculating the environmental factors related to C. buqueti using VIF, many bioclimatic variables were excluded. Additionally, during the initial modeling process, environmental factors with low contribution rates were also removed, so the remaining factors are those highly correlated with C. buqueti. A similar method was adopted in the referenced study, where 37 environmental factors were selected, resulting in the retention of 6 bioclimatic variables and 2 topographic factors. Based on our usual modeling practices, the retained environmental factors are related to the species. In future modeling work, we will aim to select more environmental factors to comprehensively assess variables related to species distribution. |
Comments 2: Lines 232-238 in Materials and Methods: Why choose 0.05 as the threshold for suitable habitat? This seems unreasonable. This method refers to research literature on Agastache rugosa, and I believe there are significant differences between this species and the species studied in this study. Why should this study be referenced? Did the model in this study run 10 times? Which one was chosen as the final result? According to the research results, it seems that the average value was chosen as the final result, and it seems that the result with the highest AUC should be selected as the final result. The introduction and discussion are not well written, and there are still many minor issues. The author seems to have not taken this manuscript seriously. Many citations in this study are inappropriate, and the author seems to not know how to cite references. The specific issues are as follows: |
Response 2: Thank you for pointing this out. The thresholds of 0.05, 0.33, and 0.66 were used to categorize suitable habitats, and this evaluation standard references the method for assessing probabilities as outlined in the IPCC's 2007 report. The study on Agastache rugosa was cited because it referenced the use of ArcGIS software to classify model results. In this study, the model was run 10 times using MaxEnt software, and the average value was ultimately chosen as the final result. According to the literature, most studies adopt the average value as the final result, as this approach helps reduce errors. Additionally, the AUC value of the model in this study reached 0.981, indicating that the model results are highly reliable. Considering these two points, I believe that choosing the average value is more appropriate. |
Comments 3: Title: The “Cyrotrachelus buqueti Guer” should be changed to “Cyrotrachelus buqueti Guer (Coleoptera: Curculionidae)”. |
Response 3: Thank you for pointing this out. Title has been revised; please see the manuscript for details.First page, first line. |
Comments 4: Line 7 in page 1: The word is repeated. |
Response 4: Thank you for pointing this out. This error has been corrected; please see page 1, line 7 of the manuscript for details. |
Comments 5: Line 10 in page 1: This content is repetitive with the content on line 6. |
Response 5: Thank you for pointing this out. This error has been corrected; please see page 1, line 10 of the manuscript for details. |
Comments 6: Lines 20-21 in page 1: It needs to be explained which period the result is, I think it is the result of the current period. |
Response 6: Thank you for pointing this out. It represents the results for the current period. The semantic ambiguity caused by the issue with my narrative order has been corrected.Lines 16-17 in page 1. |
Comments 7: Lines 31-32: The “Cyrotrachelus buqueti” and “Cyrotrachelus” need to be italicized. |
Response 7: Thank you for pointing this out. This error has been corrected; please see page 2, line 32-33 of the manuscript for details. |
Comments 8: Lines 47-50, 50-53, 53-56, 107-109, 109-112: The citation should only include the surname, and the citation should be after the person's name rather than at the end of the sentence.For example, “Jiawen Wang et al. analyzed the geographical distribution and main occurrence areas of C. buqueti in Sichuan Province, China, and found that the southeastern region of Sichuan Province is the most severely affected area by this pest[3].” should be changed to “Wang et al. [3] analyzed the geographical distribution and main occurrence areas of C. buqueti in Sichuan Province, China, and found that the southeastern region of Sichuan Province is the most severely affected area by this pest.”. |
Response 8: Thank you for pointing this out. This error has been corrected; please see page 2, line 47-56,page 6,line 106-110 of the manuscript for details. |
Comments 9: Lines 59-61, 61-63, 65-67, 67-68, 68-69, 84-86, 93-94, 94-98, 98-100, 100-103, 103-105, 207: These contents require citations. |
Response 9: Thank you for pointing this out. This error has been corrected; please see page 2, line 59-69 ,page 3,line 85-86 ,line 93-106 ,and page 5, line 187 of the manuscript for details. |
Comments 10: Lines 77-118: This paragraph is too long and needs to be segmented, and there is too much introduction about the MaxEnt model, which needs to be streamlined. |
Response 10: Thank you for pointing this out. This error has been corrected; please see page 3, line 76-109 of the manuscript for details. |
Comments 11: Lines 82-84, 87-90: These conclusions require citations, and this content overlaps with the content on lines 90-93. |
Response 11: Thank you for pointing this out. This error has been corrected; please see page 3, line 85-91 of the manuscript for details. |
Comments 12: Lines 86-87: This sentence seems unrelated to the content of this paragraph, it is recommended to delete it. |
Response 12: Thank you for pointing this out. I have deleted it. |
Comments 13: Lines 90-93: Each point of these contents needs to be accompanied by a citation, not just at the end of the sentence. |
Response 13: Thank you for pointing this out. This error has been corrected; please see page 3, line 87-92 of the manuscript for details. |
Comments 14: Lines 107-112: Why are these references cited? They seem useless and it is recommended to delete them. |
Response 14: Thank you for pointing this out. This error has been corrected; please see page 3, line 104-109 of the manuscript for details. |
Comments 15: Lines 113-114: What are the challenges? Is that the content of the next sentence? These are not clearly stated and lack citations. |
Response 15: Thank you for pointing this out. This error has been corrected; please see page 3, line 104-109 of the manuscript for details. |
Comments 16: Lines 122-132: The format and capitalization of these contents need to be checked, and many of them are incorrect. |
Response 16: Thank you for pointing this out. This error has been corrected; please see page 3, line 114-122 of the manuscript for details. |
Comments 17: Line 127: This study does not seem to include soil type data, but rather terrain data. |
Response 17: Thank you for pointing this out. This error has been corrected; please see page 3, line 119 of the manuscript for details. |
Comments 18: Line 136: What literature did you obtain the distribution data of C. buqueti from? These literature need to be cited. Did you obtain distribution data of C. buqueti from this literature? This literature seems unrelated to C. buqueti. According to Data availability, it seems that only distribution data from GBIF has been obtained, and the data source should be explained truthfully, not made up. |
Response 18: Thank you for pointing this out. This error has been corrected; please see page 3-4, line 128-139 of the manuscript for details. The primary source of the data has been cited and made publicly available on the Figshare website. |
Comments 19: Lines 136, 150, 153: The “website:” should be deleted. These links in the manuscript need to add access times, which can be referenced to the Data availability. |
Response 19: Thank you for pointing this out. This error has been corrected; please see page 3-4, line 129, 145 and 149 of the manuscript for details. |
Comments 20: Lines 143, 151, 171, 173, 207, 210, 222: These software need to provide citations or links or information about the company, country, etc. they belong to. For example, “ArcGIS 10.8” can be changed to “ArcGIS v10.8 (ESRI Inc., CA, USA)”. Additionally, these software require corresponding version numbers to be provided. |
Response 20: Thank you for pointing this out. This error has been corrected; please see page 4,5,6 line 137, 146, 167,169, 189, 192, and 205 of the manuscript for details. |
Comments 21: Lines 145-146: The occurrence records of C. buqueti need to be provided and indicate which ones come from GBIF and which ones come from literature. |
Response 21: Thank you for pointing this out. The primary source of the data has been cited and made publicly available on the Figshare website. |
Comments 22: Line 148: The resolution of bioclimatic data needs to be stated. |
Response 22: Thank you for pointing this out. This error has been corrected on page 4 line 142. |
Comments 23: Lines 178-179, 247-248, 288, 301: The format is incorrect, and the “(Table 2)” should be changed to “(Table 2)”. Lines 181-196: These seem to need to be explained in the introduction, and only a brief introduction is needed in the materials and methods. It is suggested to delete or condense them into 1-2 sentences. |
Response 23: Thank you for pointing this out. This error has been corrected on page 5, line 175; page 6, line 230; page 8, line 265; page 11, line 283; |
Comments 24: Lines 208-210: This citation is incorrect, and it is not the article that developed this software package. |
Response 24: Thank you for pointing this out. This error has been corrected on page 5, line 189. |
Comments 25: Line 224: The title is not accurate, as this section includes the establishment and evaluation of the model. |
Response 25: Thank you for pointing this out. This error has been corrected on page 6, line 207. |
Comments 26: Line 226: The link is incorrect, the software cannot be found. |
Response 26: Thank you for pointing this out. This error has been corrected on page 6, line 204. |
Comments 27: Lines 244-246: This sentence seems redundant, it is recommended to delete it. |
Response 27: Thank you for pointing this out. I have deleted it. |
Comments 28: Lines 250-259: The format of the table have issues and need to be modified. |
Response 28: Thank you for pointing this out. This error has been corrected on page 6, line 233. |
Comments 29: Lines 273-274: The Latin name in the figure is written incorrectly, 'Guer' does not need to be italicized |
Response 29: Thank you for pointing this out. This error has been corrected on page 7, line 235-247. |
Comments 30: Lines 275-288: The format of the title needs to be modified. The expression of the content is seriously incorrect. The result of this paragraph is the location of the suitable area, not the distribution location of C. buqueti. These regions are only areas where C. buqueti may be distributed, but they may not necessarily have its distribution. |
Response 30: Thank you for pointing this out. This error has been corrected on page 8, line 250-266. |
Comments 31: Lines 291-301: The description of this section is insufficient, and it is necessary to clarify the approximate location of the suitable area for each period, as well as the differences from the current period, rather than simply discussing changes in the area of suitable areas. |
Response 31: Thank you for pointing this out. This error has been corrected on page 9, line 269-276. |
Comments 32: Lines 303-305: The annotation of the figure needs clarification, as there is no 'A', 'B', 'C', 'D', 'E', or 'F' in the figure. The figure is small and not easy to observe. |
Response 32: Thank you for pointing this out. This error has been corrected on page 10, line 284-287. |
Comments 33: From here on, the line numbers are discontinuous from the previous text. |
Response 33: Thank you for pointing this out. This error has been corrected on page 12, line 288. |
Comments 34: Line 19: This sentence is a bit confusing, I think maybe “The predictions from the MaxEnt model vary under different climate scenarios.” should be changed to “The prediction results of MaxEnt model vary under different climate scenarios.”. |
Response 34: Thank you for pointing this out. This error has been corrected on page 12, line 307. |
Comments 35: Lines 25-26: This sentence seems unrelated to the content of this paragraph, it is recommended to delete it. |
Response 35: Thank you for pointing this out. I have deleted it. |
Comments 36: Lines 27-29: There is only one citation here, which does not seem to support the conclusion that the model is widely used. |
Response 36: Thank you for pointing this out. I have deleted it. |
Comments 37: Line 43: The "the average temperature during the wettest quarter" should use the standard name "mean temperature of wettest quarter", which can be found in: https://www.worldclim.org/data/bioclim.html. These variables have standard names and cannot be changed. Other areas that need to be modified need to be modified by the author themselves and will not be indicated. |
Response 37: Thank you for pointing this out. This error has been corrected on page 13, line 328-330. |
Comments 38: Lines 43-44: The citation should only include the surname, and the citation should be after the person's name rather than at the end of the sentence. |
Response 38: Thank you for pointing this out. This error has been corrected on page 13, line 327. |
Comments 39: Lines 51-64: These contents simply describe the research results, which have already been explained within the results and should not be repeated. The reasons for these phenomena should be explained, but the author did not explain them at all. |
Response 39: Thank you for pointing this out. This error has been corrected on page 13, line 334-346. |
Comments 40: Lines 53-54: This study does not support this conclusion. |
Response 40: Thank you for pointing this out. I have deleted it. |
Comments 41: Lines 65-98: This paragraph is too long and needs to be segmented. This paragraph needs to be streamlined. Although the control of this species is important, these methods are irrelevant to the conclusions of this study. These contents can explain, but they should not occupy such a large space. Specific prevention and control plans should be proposed based on this study. |
Response 41: Thank you for pointing this out. This section has been revised on page 13-14, line 347-369. |
Comments 42: Lines 102-105, 112-115: These visions are great, but has this study solved these problems? The author needs to carefully consider and revise the manuscript to achieve these goals. |
Response 41: Thank you for pointing this out. I believe this research can help address these issues. The content of this study can assist relevant authorities in preventing *C. buqueti* in non-invaded areas and in stopping its spread in invaded areas. However, since these issues are quite large and complex, they require collaboration among many researchers to develop final solutions. I think the reviewers' perspectives are crucial for the future prevention of harmful insects, so we will incorporate these considerations and research into our future studies. |
Reviewer 2 Report
Comments and Suggestions for Authors
This manuscript presents a model of the pest species Cyrtotrachelus buqueti, projected to a global distribution. The authors argue that climate change could lead to an expansion in high suitability areas for the species, and provide maps to guide managers in detecting the potential current and future distribution of the species. The manuscript is well-written and flows well, however, there are some methodological concerns, especially related to the citation of occurrence points, and the treatment of the study area.
General:
I think it is necessary to include a map of the study area including datapoints used in modeling. From a search on GBIF for the species, there are discrepancies with the scientific name – one version comes up with occurrences all over the world, another comes up with occurrences in only China and Thailand, and the table of occurrences shows a variety of scientific names and families.
I also have a concern that the authors do not specify the study extent, and that they may have used areas within that extent that are not accessible to the species. This is problematic because background points dropped in areas where the species could find suitable habitat but are not there because of dispersal can underestimate the suitability of a global model especially.
Important question for intro/discussion: Besides bamboo, does Cyrtotrachelus buqueti parasitize other species of plants? Since bamboo does not occur everywhere in the world, what bearing does the invasion of C. buqueti have if its host species is not present? Also, it is likely that the distribution of bamboo has a huge effect on the distribution of C. buqueti, which may be confounding the model.
Specific comments:
Make sure C. buqueti is italicized throughout.
Paragraph 2 of introduction: the first 4 sentences start with the word “climate.” Try to reword these sentences using different structures or transition words.
Since this research focuses on the potential global distribution of this pest, I think it would be worth mentioning the potential pathways which could lead to it becoming invasive elsewhere. Also, where is the native range of this species, and where has it been introduced/spreading?
In the introduction when you write about MaxEnt and its advantages, it is also worth noting that it does not require absence points, which can be costly/difficult to get thorough datasets on.
Line 136: When citing the GBIF dataset, you must include the full citation with DOI (otherwise they will email you complaining you did not cite them correctly 😊)
Line 148-150: The WorldClim dataset should be cited as:
Fick, S.E. and R.J. Hijmans, 2017. WorldClim 2: new 1km spatial resolution climate surfaces for global land areas. International Journal of Climatology 37 (12): 4302-4315.
Line 163: Rather than starting this sentence with “These environmental variables are used to model the ecological niche environment for species and serve as the foundation for building species distribution models.” try “Nineteen bioclimatic and three terrain variables (Table 1) were considered when building the models for C. buqueti.”
Line 173-175: How were highly correlated variables removed? Typically the protocol involves testing the variable contribution/importance in a preliminary model or using jackknife to identify important variables, then removing ones that are correlated but not contributing as much to the model.
Line 181-196: Since you already describe MaxEnt in the introduction and mention it elsewhere in your methods, it is not necessary to include this subsection. If you want, work some of it into the introduction if the information is new.
Line 197: Please state which cross-validation type was used for model tuning with ENMeval (if used).
Line 200: I believe FC refers to feature class, not feature combination
Line 233-238: What are the numbers used to reclassify based on? Are they specific thresholds from the maxent output, or are they based on previous research?
Line 242: Replace “the figure 1 below” with “the averaged model.” Also, include the standard deviation of AUC across models.
Table 3: Typo – should be “elev” not “elve”
Line 275-276: This subsection heading seems to have a formatting error.
Line 278-279: Be careful when you state that the species is distributed across Oceania, etc. Rather, the suitable area is distributed across these regions as indicated by modeling. Same for the discussion section – make sure you specify this is predicted, not necessarily actual distribution, since according to GBIF it does not appear the species occurs in the northeast US, for example.
Figure 8: While this is an interesting map, I think taking a global centroid is not as meaningful as taking the centroid of a smaller region, especially since the species does not have any occurrences, nor is modeled to be distributed anywhere around the area shown in the map. This is perhaps better suited for a table or in the text as change in degrees north, south, east, or west.
Author Response
Comments 1: This manuscript presents a model of the pest species Cyrtotrachelus buqueti, projected to a global distribution. The authors argue that climate change could lead to an expansion in high suitability areas for the species, and provide maps to guide managers in detecting the potential current and future distribution of the species. The manuscript is well-written and flows well, however, there are some methodological concerns, especially related to the citation of occurrence points, and the treatment of the study area. General: I think it is necessary to include a map of the study area including datapoints used in modeling. From a search on GBIF for the species, there are discrepancies with the scientific name – one version comes up with occurrences all over the world, another comes up with occurrences in only China and Thailand, and the table of occurrences shows a variety of scientific names and families. I also have a concern that the authors do not specify the study extent, and that they may have used areas within that extent that are not accessible to the species. This is problematic because background points dropped in areas where the species could find suitable habitat but are not there because of dispersal can underestimate the suitability of a global model especially. Important question for intro/discussion: Besides bamboo, does Cyrtotrachelus buqueti parasitize other species of plants? Since bamboo does not occur everywhere in the world, what bearing does the invasion of C. buqueti have if its host species is not present? Also, it is likely that the distribution of bamboo has a huge effect on the distribution of C. buqueti, which may be confounding the model. |
Response 1: Thank you for pointing this out. The manuscript has corrected the scientific name of the species. There is limited species distribution information available on GBIF, and most of our data come from literature sources, with the data having been made publicly available on the Figshare website. Current research shows that Cyrtotrachelus buqueti primarily parasitizes bamboo, and it is not yet clear whether it can survive on other hosts. Therefore, we cannot conclude that this species is unsuitable for areas without bamboo. Thus, the focus of this study remains on the potential distribution of suitable habitats for C. buqueti. The reviewers' comments have provided new insights into exploring the distribution of suitable habitats for C. buqueti, and we will consider host distribution more thoroughly in future research. |
Comments 2: Make sure C. buqueti is italicized throughout. |
Response 2: Thank you for pointing this out. This error has been corrected; please see the manuscript for details. |
Comments 3: Paragraph 2 of introduction: the first 4 sentences start with the word “climate.” Try to reword these sentences using different structures or transition words. |
Response 3: Thank you for pointing this out. This error has been corrected; please see page 2, line 60 of the manuscript for details. |
Comments 4: Since this research focuses on the potential global distribution of this pest, I think it would be worth mentioning the potential pathways which could lead to it becoming invasive elsewhere. Also, where is the native range of this species, and where has it been introduced/spreading? |
Response 4: Thank you for pointing this out. There is limited research on the native range and origin of C. buqueti. This study explored the current major distribution areas of this species, but there is little literature related to its introduction and spread. In future research, we will focus more on the origin and dissemination of the species. |
Comments 5: In the introduction when you write about MaxEnt and its advantages, it is also worth noting that it does not require absence points, which can be costly/difficult to get thorough datasets on. |
Response 5: Thank you for pointing this out. I've added this on page 3 line 90-91. |
Comments 6: Line 136: When citing the GBIF dataset, you must include the full citation with DOI (otherwise they will email you complaining you did not cite them correctly) |
Response 6: Thank you for pointing this out.The full citation and DOI are indicated in the Data availability section at the end of the article. |
Comments 7: Line 148-150: The WorldClim dataset should be cited as: Fick, S.E. and R.J. Hijmans, 2017. WorldClim 2: new 1km spatial resolution climate surfaces for global land areas. International Journal of Climatology 37 (12): 4302-4315. |
Response 7: Thank you for pointing this out. This error has been corrected; please see page 4, line 145 of the manuscript for details. |
Comments 8: Line 163: Rather than starting this sentence with “These environmental variables are used to model the ecological niche environment for species and serve as the foundation for building species distribution models.” try “Nineteen bioclimatic and three terrain variables (Table 1) were considered when building the models for C. buqueti.” |
Response 8: Thank you for pointing this out. This error has been corrected; please see page 5, line 159 of the manuscript for details. |
Comments 9: Line 173-175: How were highly correlated variables removed? Typically the protocol involves testing the variable contribution/importance in a preliminary model or using jackknife to identify important variables, then removing ones that are correlated but not contributing as much to the model. |
Response 9: Yes, we have adopted these methods with the addition of VIF to screen for environmental factors with multicollinearity, thus reducing the degree of fit. |
Comments 10: Line 181-196: Since you already describe MaxEnt in the introduction and mention it elsewhere in your methods, it is not necessary to include this subsection. If you want, work some of it into the introduction if the information is new. |
Response 10: Thank you for pointing this out. I have deleted it. |
Comments 11: Line 197: Please state which cross-validation type was used for model tuning with ENMeval (if used). |
Response 11: Sorry, I don't know what this message refers to. |
Comments 12: I believe FC refers to feature class, not feature combination |
Response 12: Thank you for pointing this out.Many papers use FC to represent feature combination, so in this paper, FC also represents feature combination. Since different types of features can be combined in the MaxEnt model construction, I think using feature combination is more appropriate. |
Comments 13: Line 233-238: What are the numbers used to reclassify based on? Are they specific thresholds from the maxent output, or are they based on previous research? |
Response 13: Thank you for pointing this out. This assessment standard references the IPCC's method of dividing up the assessment of likelihood in its 2007 report. |
Comments 14: Line 242: Replace “the figure 1 below” with “the averaged model.” Also, include the standard deviation of AUC across models. |
Response 14: Thank you for pointing this out. This error has been corrected; please see page 6, line 225 of the manuscript for details. |
Comments 15: Table 3: Typo – should be “elev” not “elve” |
Response 15: Thank you for pointing this out. This error has been corrected; please see page 6, line 233 of the manuscript for details. |
Comments 16: Line 275-276: This subsection heading seems to have a formatting error. |
Response 16: Thank you for pointing this out. This error has been corrected; please see page 8, line 250 of the manuscript for details. |
Comments 17: Line 278-279: Be careful when you state that the species is distributed across Oceania, etc. Rather, the suitable area is distributed across these regions as indicated by modeling. Same for the discussion section – make sure you specify this is predicted, not necessarily actual distribution, since according to GBIF it does not appear the species occurs in the northeast US, for example. |
Response 17: Thank you for pointing this out.This error has been corrected; please see page 8, line 250-266 of the manuscript for details. |
Comments 18: Figure 8: While this is an interesting map, I think taking a global centroid is not as meaningful as taking the centroid of a smaller region, especially since the species does not have any occurrences, nor is modeled to be distributed anywhere around the area shown in the map. This is perhaps better suited for a table or in the text as change in degrees north, south, east, or west. |
Response 18: Thank you for pointing this out. Because this species is mainly distributed in China, but its center of mass in China has already been studied. However, the global center of mass has not been studied yet, so we think it is worth exploring the changes in its global center of mass to obtain trends in its future distribution of center of mass. |

Reviewer 3 Report
Comments and Suggestions for Authors
The title needs to be corrected to:
Global Distribution Prediction of Cyrtotrachelus buqueti Guérin-Méneville, 1844 (Coleoptera: Dryophthoridae)
Insights from the Optimised MaxEnt Model
The text contains easily correctable technical errors.
Overall, the article touches upon the important issue of changes in the pest's range due to global warming.
13: MaxEnt
16: must be space before km2
20: must be space before km2
21-22: precipitation of the driest month (Bio14), mean temperature of the wettest quarter (Bio08), and altitude (Elev).
31: Cyrtotrachelus buqueti - cursive
31-32: delete "Guer (Coleoptera: Curculionidae) is a species of beetle in the genus Cyrtotrachelus within the family Curculionidae. It"
33: Bambusa, Dendrocalamopsis, and Dendrocalamus cursive
33, 37, 40,44, 48, 53-54,55,: C. buqueti - cursive
46: add India
108: Oncomelania hupensis - cursive
180: Must be dot after Table 2
250: Must be dot after Table 3
Delete Guer in fig. 2.
262: Must be dot after Fig 2
Delete Guer in fig. 3 and fig. 4 .
In fig. 4:
change bio08 to mean temperature of the wettest quarter
change elev to altitude
change bio14 to precipitation of the driest month
276: current climatic conditions - cursive
287-288: must be space before km2
Title of table 4: must be dot after Table 4.
Table 4: must be space before km2, must be km, not Km
Page 12, line 16: Must be dot after Fig. 8
Page 13, line 29: Euplatypus parallelus Bright & Skidmore, 2002
Page 13, line 52-53: must be space before km2
Page 13, line 86 Beauveria brongniartii (Sacc.) Petch.

Author Response
Comments 1: The title needs to be corrected to: Global Distribution Prediction of Cyrtotrachelus buqueti Guérin-Méneville, 1844 (Coleoptera: Dryophthoridae) Insights from the Optimised MaxEnt Model |
Response 1: Thank you for pointing this out. Title has been revised; please see the manuscript for details.First page, first line. |
Comments 2: 13: MaxEnt |
Response 2: Thank you for pointing this out. I have added it on page 1 line 13. |
Comments 3: 16 must be space before km2 |
Response 3:Thank you for pointing this out. This error has been corrected; please see page 1, line 19 of the manuscript for details. |
Comments 4:20 must be space before km2 |
Response 4:Thank you for pointing this out. This error has been corrected; please see page 1, line 22 of the manuscript for details. |
Comments 5: 21-22: precipitation of the driest month (Bio14), mean temperature of the wettest quarter (Bio08), and altitude (Elev). |
Response 5: Thank you for pointing this out. This error has been corrected; please see page 1, line 17-18 of the manuscript for details. |
Comments 6: 31: Cyrtotrachelus buqueti - cursive |
Response 6:Thank you for pointing this out. This error has been corrected; please see page 2, line 32 of the manuscript for details. |
Comments 7: 31-32: delete "Guer (Coleoptera: Curculionidae) is a species of beetle in the genus Cyrtotrachelus within the family Curculionidae. It" |
Response 7:Thank you for pointing this out. This error has been corrected; please see page 2, line 32 of the manuscript for details. |
Comments 8: 33: Bambusa, Dendrocalamopsis, and Dendrocalamus cursive |
Response 8:Thank you for pointing this out. This error has been corrected; please see page 2, line 32-33 of the manuscript for details. |
Comments 9: 33, 37, 40,44, 48, 53-54,55,: C. buqueti - cursive |
Response 9:Thank you for pointing this out. This error has been corrected; please see the manuscript for details. |
Comments 10: 46: add India |
Response 10:Thank you for pointing this out. This error has been corrected; please see page 2, line 47 of the manuscript for details. |
Comments 11: 108: Oncomelania hupensis - cursive |
Response 11:Thank you for pointing this out. This error has been corrected; please see page 3, line 107 of the manuscript for details. |
Comments 12: 180: Must be dot after Table 2 |
Response 12:Thank you for pointing this out. This error has been corrected; please see page 5, line 117 of the manuscript for details. |
Comments 13: 250: Must be dot after Table 3 |
Response 13:Thank you for pointing this out. This error has been corrected; please see page 6, line 233 of the manuscript for details. |
Comments 14: Delete Guer in fig. 2. |
Response 14:Thank you for pointing this out. This error has been corrected; please see page 7, line 244 of the manuscript for details. |
Comments 15: 262: Must be dot after Fig 2 |
Response 15:Thank you for pointing this out. This error has been corrected; please see page 7, line 244 of the manuscript for details. |
Comments 16: Delete Guer in fig. 3 and fig. 4 . |
Response 16:Thank you for pointing this out. This error has been corrected; please see page 8, line 248 of the manuscript for details. |
Comments 17: In fig. 4: change bio08 to mean temperature of the wettest quarter change elev to altitude change bio14 to precipitation of the driest month |
Response 17:Thank you for pointing this out. I think these terms have already been introduced in the article, and it would be superfluous to show them in detail in the figure. |
Comments 18: 276: current climatic conditions - cursive |
Response 15:Thank you for pointing this out. This error has been corrected; please see page 8, line 250 of the manuscript for details. |
Comments 19: 287-288: must be space before km2 |
Response 19:Thank you for pointing this out. This error has been corrected; please see page 9, line 279-283 of the manuscript for details. |
Comments 20: Title of table 4: must be dot after Table 4. |
Response 20:Thank you for pointing this out. This error has been corrected; please see page 11, line 287 of the manuscript for details. |
Comments 21: Table 4: must be space before km2, must be km, not Km |
Response 21:Thank you for pointing this out. This error has been corrected; please see page 11, line 287 of the manuscript for details. |
Comments 22: Page 12, line 16: Must be dot after Fig. 8 |
Response 22:Thank you for pointing this out. This error has been corrected; please see page 12, line 303 of the manuscript for details. |
Comments 23: Page 13, line 29: Euplatypus parallelus Bright & Skidmore, 2002 |
Response 23:Response 22:Thank you for pointing this out. I have deleted it. |
Comments 24: Page 13, line 52-53: must be space before km2 |
Response 24:Thank you for pointing this out. This error has been corrected; please see page 13, line 334-337 of the manuscript for details. |
Comments 25: Page 13, line 86 Beauveria brongniartii (Sacc.) Petch. |
Response 25:Thank you for pointing this out. This error has been corrected; please see page 13, line 360 of the manuscript for details. |

Round 2
Reviewer 1 Report
Comments and Suggestions for Authors
This manuscript has not been carefully revised, only minor issues have been addressed, but there are still many minor problems within the manuscript. It is extremely surprising that some of the issues raised in the previous round of review were claimed to have been revised by the author, but in fact, they were not revised. Furthermore, it is particularly important that the two key issues I raised in the previous round have not been resolved, or rather, the author did not intend to address them. If these issues are not resolved, this manuscript should not be accepted.
1. The method of selecting environmental factors still lacks basis and is absurd. In lines 154-156, the author states that selecting environmental factors is to exclude variables that do not contribute significantly to the prediction, so why exclude those variables that are correlated? Moreover, since VIF can predict the correlation between variables, why conduct Pearson correlation analysis? The author's explanation is contradictory.
Additionally, it should be noted that the highly correlated environmental variables is not about excluding them, but about selecting one that is more important. It is extremely absurd for the author to exclude all correlated variables. These were already explained in the previous round of review, but the author turned a blind eye to this comment and refused to change it, and the explanation given was also difficult to convince.
How the author chooses more environmental factors for future research is irrelevant to this study, and key variables that affect the distribution of Cyrtotrachelus buqueti are likely to be excluded in this study. The research method used in this study is unacceptable, and the credibility of the research results is also questionable. The author must re select environmental factors and re model, otherwise the research results are unreliable and this manuscript is not suitable for acceptance.
2. Since this manuscript refers to the methods in the IPCC's 2007 report, why not directly cite the report and instead cite the literature on Agastache rugosa? It is puzzling that this study did not use this method, which is too bizarre and inexplicable.
In addition, the author needs to point out where this method comes from in the IPCC's 2007 report, as it does not seem to exist in the report (https://www.ipcc.ch/site/assets/uploads/2018/06/ar4_syr.pdf). The selection of the threshold for the suitable habitat is the cornerstone of this study, and if the threshold is incorrect, the results of the study will not be convincing.
3. The discussion was not well written, and some issues were already pointed out in the previous round of review, but the author did not revise them. These contents in lines 334-346 simply describe the research results, which have already been explained within the results and should not be repeated. The reasons for these phenomena should be explained, but the author did not explain them at all.
These contents in lines 347-368 can explain, but they should not occupy such a large space. Although the control of this species is important, these methods are irrelevant to the conclusions of this study. Specific prevention and control plans should be proposed based on this study.
The introduction and conclusion sections of this article elaborate on many research objectives and significance, although many questions can be explained, this study does not explain or achieve these objectives at all. The author did not explain the changes in the suitable distribution area of C. buqueti under future climate change, the reasons for these changes, and the role of these changes in the prevention and management of the species. If these key issues are not explained, are the purposes and meanings claimed in this article just for readers to imagine?
4. The conclusion section is not well written. The conclusion should be based on your research objectives and explain what conclusions have been drawn from this study. The content of lines 374, 382-385 needs to be modified or deleted, as they are meaningless. The significance of this study should be elaborated in detail, rather than constantly stating these vague words. The purpose and significance of this study cannot be imagined solely by readers or require readers to further analyze. These comments were already pointed out in the previous review, with the aim of encouraging the author to make revisions, rather than listening to you say that you will complete them in future research. And lines 371-372 are not the conclusion of this study.
There are still some minor issues that need to be addressed, the specific issues are as follows:
5. Lines 2-3: This Latin name seems to be incorrect.
6. Line 13: The “Maximum Entropy Model (Maxent)” should be changed to “Maximum Entropy (Maxent) Model”.
7. Line 31: The expression “Cyrotrachelus buqueti or Long Armed Snout Beetle” seems inappropriate, is “Long Armed Snout Beetle” alternative name of C. buqueti? It's hard to tell that “Long Armed Snout Beetle” is its alternative name in this way of expression.
8. Line 32: The “in-sect” should be changed to “insect”.
9. Line 42: This citation seems to be incorrect, why is this reference cited? Why not cite the Catalogue of Forest Pests in China? The author should quote correctly and not quote arbitrarily. In the previous round of review, many citation issues were already pointed out, and I am unable to assist the author in verifying the correctness of each citation individually. This requires the author to verify on their own.
10. Lines 46-47: These contents seem unrelated to this study, and I don't know why the author cited this part of the content.
11. Lines 56-61: These contents are all elaborating on global warming. Although global warming is a significant feature of future climate change, this phenomenon cannot represent future climate change. Future climate change is complex and diverse, not just global warming.
12. Line 69: These contents require citations.
13. Line 88: The meaning of this sentence is difficult to understand.
14. Lines 104-107: Why are these two references cited? These contents seem redundant.
15. Lines 118-120: This research objective seems to overlap with the first research objective.
16. Line 125: This literature seems unrelated to C. buqueti. Did you obtain distribution data of C. buqueti from this literature? What literature did you obtain the distribution data of C. buqueti from? These literature need to be cited. The author seems to not know how to cite references. Which was pointed out in the previous round of review. It is extremely surprising that some of the issues raised in the previous round of review were claimed to have been revised by the author, but in fact, they were not revised.
17. Line 135: What does the citation mean here? Did you use the distribution data from these two references? It should not be quoted in this way.
18. Line 181: The citation does not match the reference, and the author seems to have made a mistake with their last name and first name.
19. Line 196: The reference or link of this software is incorrect.
20. Lines 216-239: It needs to be explained which period the result is, I think it is the result of the current period.
21. 3.3. Potential habitat changes of C. buqueti under future climate scenarios: The description of this section is insufficient, and it is necessary to clarify the differences from the current period, rather than simply discussing changes in the area of suitable areas. Which was pointed out in the previous round of review. The author claims to have made revisions, but the changes are minimal.
22. 3.4. Centroid changes in potential distribution: The content of this section seems meaningless because the author hardly explains this part in the discussion and conclusion sections. What is the significance of this section? What role does it play in this study? If these are not explained, this section should be deleted.
23. Line 307: This sentence seems unrelated to the following content.
24. Line 321: Are these the research results of the two cited literature? But it seems to be the research result of this study, so why are these two references cited here?
25. Line 325: The “emergence rate” should be changed to “probability of occurrence”. This study does not seem to investigate the emergence rate of C. buqueti.
26. Lines 324-346: These contents simply describe the research results, which have already been explained within the results and should not be repeated. The reasons for these phenomena should be explained, but the author did not explain them at all. Which was pointed out in the previous round of review. It is extremely surprising that some of the issues raised in the previous round of review were claimed to have been revised by the author, but in fact, they were not revised.
27. Lines 342-344: In lines 211-213, this study divides potential habitats into highly suitable habitat (0.66 ≤ P ≤ 1), moderately suitable habitat (0.33 ≤ P < 0.66), marginally suitable habitat (0.05 ≤ P < 0.33), and unsuitable habitat (P < 0.05).
But here the author refers to it as low-suitability areas, moderate-suitability areas, and high-suitability areas. The classification of potential habitats into which types should be unified throughout this study, rather than being arbitrarily divided. Whether this issue exists elsewhere in this manuscript needs to be verified by the author themselves, so I will not elaborate further.
28. Lines 347-368: Although the control of this species is important, these methods are irrelevant to the conclusions of this study. These contents can explain, but they should not occupy such a large space. Specific prevention and control plans should be proposed based on this study. Which was pointed out in the previous round of review. It is extremely surprising that some of the issues raised in the previous round of review were claimed to have been revised by the author, but in fact, they were not revised.
29. Lines 359: The link seems inaccessible. These distribution data are recommended to be published as supporting materials for this study, in addition to being uploaded on figshare.
30. Figures 3, 4, and 5 are not clear and difficult to observe.
Comments on the Quality of English LanguageI have explained the specific issues in the comments.
Author Response
Dear editors and reviewers,
We are very grateful for your constructive comments and suggestions for our manuscript entitled "Global Distribution Prediction of Cyrtotrachelus buqueti Guer: Insights from the Optimised MaxEnt Model" (Manuscript ID: insects-3162695). Your comments are very valuable and helpful for improving our manuscript. In the following, the responses to all the comments are provided one by one. The revisions have been marked in blue font in the manuscript.
We have tried our best to make all the revisions clear, and we hope that the revised manuscript can satisfy the requirements for publication. The main corrections in the paper and the responds to the reviewer's comments are as flowing:
Comments 1: 1. The method of selecting environmental factors still lacks basis and is absurd. In lines 154-156, the author states that selecting environmental factors is to exclude variables that do not contribute significantly to the prediction, so why exclude those variables that are correlated? Moreover, since VIF can predict the correlation between variables, why conduct Pearson correlation analysis? The author's explanation is contradictory. Additionally, it should be noted that the highly correlated environmental variables is not about excluding them, but about selecting one that is more important. It is extremely absurd for the author to exclude all correlated variables. These were already explained in the previous round of review, but the author turned a blind eye to this comment and refused to change it, and the explanation given was also difficult to convince. How the author chooses more environmental factors for future research is irrelevant to this study, and key variables that affect the distribution of Cyrtotrachelus buqueti are likely to be excluded in this study. The research method used in this study is unacceptable, and the credibility of the research results is also questionable. The author must re select environmental factors and re model, otherwise the research results are unreliable and this manuscript is not suitable for acceptance. |
Response 1: Thank you for pointing out the deficiencies in this paper. When constructing the model, the environmental factors with small contributions were excluded first, and then the environmental factors with Pearson coefficients greater than 0.8 were screened out. The exclusions were irrelevant environmental factors, and I apologize for any misunderstanding my expression may have caused, I have revised the article. (Line 161-172) |
Comments 2: Since this manuscript refers to the methods in the IPCC's 2007 report, why not directly cite the report and instead cite the literature on Agastache rugosa? It is puzzling that this study did not use this method, which is too bizarre and inexplicable. In addition, the author needs to point out where this method comes from in the IPCC's 2007 report, as it does not seem to exist in the report (https://www.ipcc.ch/site/assets/uploads/2018/06/ar4_syr.pdf). The selection of the threshold for the suitable habitat is the cornerstone of this study, and if the threshold is incorrect, the results of the study will not be convincing. |
Response 2: Thank you for pointing out my error. I have corrected the error in the text and will always be mindful of this issue in my future literature citations. (Line 216-218) |
Comments 3: The discussion was not well written, and some issues were already pointed out in the previous round of review, but the author did not revise them. These contents in lines 334-346 simply describe the research results, which have already been explained within the results and should not be repeated. The reasons for these phenomena should be explained, but the author did not explain them at all. These contents in lines 347-368 can explain, but they should not occupy such a large space. Although the control of this species is important, these methods are irrelevant to the conclusions of this study. Specific prevention and control plans should be proposed based on this study. The introduction and conclusion sections of this article elaborate on many research objectives and significance, although many questions can be explained, this study does not explain or achieve these objectives at all. The author did not explain the changes in the suitable distribution area of C. buqueti under future climate change, the reasons for these changes, and the role of these changes in the prevention and management of the species. If these key issues are not explained, are the purposes and meanings claimed in this article just for readers to imagine?
|
Response 3: Thank you for pointing out my shortcomings. I have made changes to address the discussion and conclusion sections of the article. (Line 320-398) |
Comments 4: The conclusion section is not well written. The conclusion should be based on your research objectives and explain what conclusions have been drawn from this study. The content of lines 374, 382-385 needs to be modified or deleted, as they are meaningless. The significance of this study should be elaborated in detail, rather than constantly stating these vague words. The purpose and significance of this study cannot be imagined solely by readers or require readers to further analyze. These comments were already pointed out in the previous review, with the aim of encouraging the author to make revisions, rather than listening to you say that you will complete them in future research. And lines 371-372 are not the conclusion of this study. |
Response 4: Thank you for pointing out my shortcomings. I have made changes to address the discussion and conclusion sections of the article. (Line 320-398) |
Comments 5: Lines 2-3: This Latin name seems to be incorrect. |
Response 5: Thank you for pointing this out, I have made the change. (Line 2-3) |
Comments 6: Line 13: The “Maximum Entropy Model (Maxent)” should be changed to “Maximum Entropy (Maxent) Model”. |
Response 6: Thank you for pointing this out, I have made the change. (Line 13) |
Comments 7: Line 31: The expression “Cyrotrachelus buqueti or Long Armed Snout Beetle” seems inappropriate, is “Long Armed Snout Beetle” alternative name of C. buqueti? It's hard to tell that “Long Armed Snout Beetle” is its alternative name in this way of expression. |
Response 7: Thank you for pointing this out, I have made the change. (Line 33-36) |
Comments 8: Line 32: The “in-sect” should be changed to “insect”. |
Response 8: Thank you for pointing this out, I have made the change. (Line 36) |
Comments 9: Line 42: This citation seems to be incorrect, why is this reference cited? Why not cite the Catalogue of Forest Pests in China? The author should quote correctly and not quote arbitrarily. In the previous round of review, many citation issues were already pointed out, and I am unable to assist the author in verifying the correctness of each citation individually. This requires the author to verify on their own. |
Response 9: Thank you for pointing this out, I have made the change. (Line 46) |
Comments 10: Lines 46-47: These contents seem unrelated to this study, and I don't know why the author cited this part of the content. |
Response 10: Thank you for pointing this out, I have removed the content. (Line 46) |
Comments 11: Lines 56-61: These contents are all elaborating on global warming. Although global warming is a significant feature of future climate change, this phenomenon cannot represent future climate change. Future climate change is complex and diverse, not just global warming. |
Response 11: Thank you for pointing this out, I have made the change. (Line 57-59) |
Comments 12: Line 69: These contents require citations. |
Response 12: Thank you for pointing this out, I have made the change. (Line 70-72) |
Comments 13: Line 88: The meaning of this sentence is difficult to understand. |
Response 13: Thank you for pointing this out, I have made the change. (Line 85-87) |
Comments 14: Lines 104-107: Why are these two references cited? These contents seem redundant. |
Response 14: Thank you for pointing this out, I have removed the content. |
Comments 15: Lines 118-120: This research objective seems to overlap with the first research objective. |
Response 15: Thank you for pointing this out, I have removed the content. (Line 114) |
Comments 16: Line 125: This literature seems unrelated to C. buqueti. Did you obtain distribution data of C. buqueti from this literature? What literature did you obtain the distribution data of C. buqueti from? These literature need to be cited. The author seems to not know how to cite references. Which was pointed out in the previous round of review. It is extremely surprising that some of the issues raised in the previous round of review were claimed to have been revised by the author, but in fact, they were not revised. |
Response 16: Thank you for pointing this out, I have made the change. (Line 121-133) |
Comments 17: Line 135: What does the citation mean here? Did you use the distribution data from these two references? It should not be quoted in this way. |
Response 17: Thank you for pointing this out, I have made the change. (Line 133) |
Comments 18: Line 181: The citation does not match the reference, and the author seems to have made a mistake with their last name and first name. |
Response 18: Thank you for pointing this out, I have made the change. (Line 179-181) |
Comments 19: Line 196: The reference or link of this software is incorrect. |
Response 19: Thank you for pointing this out, I have made the change. (Line 200) |
Comments 20: Lines 216-239: It needs to be explained which period the result is, I think it is the result of the current period. |
Response 20: Thank you for pointing this out, I have made the change. (Line 224) |
Comments 21: 3.3. Potential habitat changes of C. buqueti under future climate scenarios: The description of this section is insufficient, and it is necessary to clarify the differences from the current period, rather than simply discussing changes in the area of suitable areas. Which was pointed out in the previous round of review. The author claims to have made revisions, but the changes are minimal. |
Response 21: Thank you for pointing this out, I have made the change. (Line 269-289) |
Comments 22: 3.4. Centroid changes in potential distribution: The content of this section seems meaningless because the author hardly explains this part in the discussion and conclusion sections. What is the significance of this section? What role does it play in this study? If these are not explained, this section should be deleted.3.4. |
Response 22: Thank you for raising this point. Exploring the changes in the center of mass of the high habitability zone of this species is primarily a clearer presentation of the direction it will move in future environments. It is a much simpler visual representation of which direction the high habitability zone of this species will shift in the future than a map of the distribution of future habitability zones. This is beneficial to the spreading area to strengthen the prevention in a directional way. In addition, I have made changes in the discussion section. (Line 358-366) |
Comments 23: Line 307: This sentence seems unrelated to the following content. |
Response 23: Thank you for pointing this out, I have made the change. (Line 307) |
Comments 24: Line 321: Are these the research results of the two cited literature? But it seems to be the research result of this study, so why are these two references cited here? |
Response 24: Thank you for pointing this out, I have made the change. (Line 317-320) |
Comments 25: Line 325: The “emergence rate” should be changed to “probability of occurrence”. This study does not seem to investigate the emergence rate of C. buqueti. |
Response 25: Thank you for pointing this out, I have made the change. |
Comments 26: Lines 324-346: These contents simply describe the research results, which have already been explained within the results and should not be repeated. The reasons for these phenomena should be explained, but the author did not explain them at all. Which was pointed out in the previous round of review. It is extremely surprising that some of the issues raised in the previous round of review were claimed to have been revised by the author, but in fact, they were not revised. |
Response 26: Thank you for pointing this out, I have made the change. (Line 321-341) |
Comments 27: Lines 342-344: In lines 211-213, this study divides potential habitats into highly suitable habitat (0.66 ≤ P ≤ 1), moderately suitable habitat (0.33 ≤ P < 0.66), marginally suitable habitat (0.05 ≤ P < 0.33), and unsuitable habitat (P < 0.05). But here the author refers to it as low-suitability areas, moderate-suitability areas, and high-suitability areas. The classification of potential habitats into which types should be unified throughout this study, rather than being arbitrarily divided. Whether this issue exists elsewhere in this manuscript needs to be verified by the author themselves, so I will not elaborate further. |
Response 27: Thank you for pointing this out, I have made the change. (Line 216-218) |
Comments 28: Lines 347-368: Although the control of this species is important, these methods are irrelevant to the conclusions of this study. These contents can explain, but they should not occupy such a large space. Specific prevention and control plans should be proposed based on this study. Which was pointed out in the previous round of review. It is extremely surprising that some of the issues raised in the previous round of review were claimed to have been revised by the author, but in fact, they were not revised. |
Response 28: Thank you for pointing this out, I have made the change. (Line 367-380) |
Comments 29: Lines 359: The link seems inaccessible. These distribution data are recommended to be published as supporting materials for this study, in addition to being uploaded on figshare. |
Response 29: Thank you for pointing this out. Can you try again to see if this link will open? (https://doi.org/10.6084/m9.figshare.26412433.v2)I was able to open it on the website, plus I added this part of the coordinate point data as Exhibit 1 to the article. (Line 125) |
Comments 30: Figures 3, 4, and 5 are not clear and difficult to observe. |
Response 30: Thank you for pointing this out. The lack of clarity may be due to compression, I have replaced all the images, if they are still not clear, please contact me. Also I have uploaded the images in 300dpi format to the zip. |
4. Response to Comments on the Quality of English Language |
Point 1:There doesn't seem to be a major issue with the language, but there are many issues with the wording, which have been explained in the comments. |
Response 1:Thank you very much to the reviewers and editors for their thorough reading and revisions of this manuscript. I will strive to enhance my research capabilities and writing skills in future work. |
5. Additional clarifications |
|
We tried our best to improve the manuscript and made some changes in the manuscript. These changes will not influence the content and framework of the paper.
We appreciate for Editors and Reviewers' warm work earnestly and hope that the correction will meet with approval.
Once again, thank you very much for your comments and suggestions.
Regards.

Reviewer 2 Report
Comments and Suggestions for Authors
Although the authors have responded to reviewer comments, I am still concerned about several points:
1. The GBIF link provided by the authors contains only 17 occurrence points contained within China and Thailand, yet the table included in the data availability statement shows significantly more occurrences from that source. This must be addressed so as to not misrepresent your methods.
2. As stated in the previous review, there should be some representation of these points on a map.
3. Additionally, the authors did not address this previous comment:
I also have a concern that the authors do not specify the study extent, and that they may have used areas within that extent that are not accessible to the species. This is problematic because background points dropped in areas where the species could find suitable habitat but are not there because of dispersal can underestimate the suitability of a global model especially.
In other words, training a model using inaccessible absence points can overestimate unsuitable area, and it is important to at least caveat that.
4. Line 135: Change "obtained" to "retained"
5. The text from ENMeval 2.0 is "regularization multiplier and feature class combinations". Even though other papers may use the term "feature combination", the source of the package which you are using for your methods does not.
Author Response
Dear editors and reviewers,
We are very grateful for your constructive comments and suggestions for our manuscript entitled "Global Distribution Prediction of Cyrtotrachelus buqueti Guer: Insights from the Optimised MaxEnt Model" (Manuscript ID: insects-3162695). Your comments are very valuable and helpful for improving our manuscript. In the following, the responses to all the comments are provided one by one. The revisions have been marked in blue font in the manuscript.
We have tried our best to make all the revisions clear, and we hope that the revised manuscript can satisfy the requirements for publication. The main corrections in the paper and the responds to the reviewer's comments are as flowing:
Comments 1: The GBIF link provided by the authors contains only 17 occurrence points contained within China and Thailand, yet the table included in the data availability statement shows significantly more occurrences from that source. This must be addressed so as to not misrepresent your methods. |
Response 1: Thank you for raising this issue. Here is where I was negligent about the reliability of my data sources. Because my data sources are mainly GBIF and another Cyrtotrachelus buqueti related paper, and some of the data in this paper are from GBIF, so I labeled some of the sources as GBIF. Some of the data in this paper came from GBIF, so I labeled some of the sources as GBIF as well.This data discrepancy may be due to a difference in the year of access, as this paper was published in 2019, so there is a discrepancy in the documentation of occurrence data for this species in GBIF. I have changed this part of the data in figshare. (Line 121-133) |
Comments 2: As stated in the previous review, there should be some representation of these points on a map. |
Response 2: Thank you for bringing this up, I apologize for missing this, I have inserted a new map with coordinate points in the article. (Line 357) |
Comments 3: Additionally, the authors did not address this previous comment: I also have a concern that the authors do not specify the study extent, and that they may have used areas within that extent that are not accessible to the species. This is problematic because background points dropped in areas where the species could find suitable habitat but are not there because of dispersal can underestimate the suitability of a global model especially. In other words, training a model using inaccessible absence points can overestimate unsuitable area, and it is important to at least caveat that. |
Response 3: Thank you for this question. What you mean is: In this study, Cyrtotrachelus buqueti's distribution is mainly in China and Thailand, so if we use this data for global modeling, it will result in a small area of global unsuitable area? This is because there are only 6 records with coordinates for this species in GBIF, and the global data are generally under-represented. However, an important feature of the MaxEnt model is its ability to generate more reliable predictions with smaller sample sizes. Therefore, after obtaining 397 data points, I think it can alleviate this problem. If you mean in relation to background points, we think that while there would be a phenomenon where the background points are within areas that are inaccessible to long-footed elephants, there would also be background points generated that would be suitable for the species in their own right. We have 10,000 randomly generated background points, which may alleviate this problem. |
Comments 4: Line 135: Change "obtained" to "retained" |
Response 4: Thank you for spotting this, I have made a change to this word. (Line 132) |
Comments 5: The text from ENMeval 2.0 is "regularization multiplier and feature class combinations". Even though other papers may use the term "feature combination", the source of the package which you are using for your methods does not. |
Response 5: Thank you for raising this issue. I have changed “feature combination” to “feature class” in the article. (Line 186,188 and 201) We tried our best to improve the manuscript and made some changes in the manuscript. These changes will not influence the content and framework of the paper. We appreciate for Editors and Reviewers' warm work earnestly and hope that the correction will meet with approval. Once again, thank you very much for your comments and suggestions. Regards.
|

Round 3
Reviewer 1 Report
Comments and Suggestions for Authors
After revision, most of the issues in this manuscript have been corrected, and the quality of the manuscript has significantly improved. However, there are still several issues that have not been completely resolved. The specific issues are as follows:
1. Lines 216-218: The method used in the literature referenced here is the natural-breaks method. If this study also used this method, it should be directly indicated. Although this method is not the best way to determine the threshold for the suitable area, it seems acceptable.
However, the classification results of the suitability index in this manuscript should not be exactly the same as the referenced literature, as the results of this study cannot be exactly the same as it. This manuscript needs to calculate the division results of suitability index in ArcGIS based on the research results.
There are still some minor issues that need to be addressed, the specific issues are as follows:
2. Line 2: The “Cyrotrachelus buqueti (Guer)” should be changed to “Cyrotrachelus buqueti Guer”.
3. Lines 33-34: The “Curculionidae” only needs to be repeated once.
4. Lines 45-46: This reference ([3]) is still incorrect. The Catalogue of Forest Pests in China needs to be cited here.
5. Line 91: The author seems to have misunderstood. The previous round of review pointed out that the third advantage of the MaxEnt model is difficult to understand, rather than the content in lines 85-87. Does the line number pointed out in my comment correspond to the line number understood by the author?
6. Lines 102-105: Why are these two references cited? These contents seem redundant. The author replied that this part of the content has been deleted, but it has not been deleted. Does the line number pointed out in my comment correspond to the line number understood by the author?
7. Line 124: The reference ([5]) cited here does not appear in Table S1.
8. Line 139: The SDM tools requires citations, you can refer to http://www.sdmtoolbox.org/technical-info.
9. Line 164: The MaxEnt software requires citations and the version number should be indicated. If it appears again in the following text, no citation is required.
10. Line 165: It should be noted here that when the contribution rate of the variable is lower than what, it will be excluded.
11. Line 166: The ENMTools.pl software requires citations.
12. Line 172: The number of variables in the title of Table 2 is incorrect.
13. Line 204: The number of variables is incorrect.
14. Lines 241-244: Is it necessary to indicate whether these results are based on the adaptability index of 0.08 or 0.49? Or indicate whether it is based on low habitat suitability or highly habitat suitability?
15. Line 267 (Figure 4): It is suggested to fully display the South China Sea in the figure on the right. Additionally, the latitude and longitude of the image are blurry.
16. Line 289: There is no need for a space at the end of this sentence.
17. Line 291 (Figure 5): The latitude and longitude of the image are blurry.
18. Line 296 (Figure 6): It is suggested to fully display the South China Sea in the figure. Additionally, the latitude and longitude of the image are blurry.
19. Line 303 (Figure 7): The latitude and longitude of the image are blurry, and the points on the right are very small, making it difficult to see what shape each point is.
20. Line 359 (Figure 8): The distribution points are blurry, and many of them cannot be seen at all, and the latitude and longitude are also blurry. It is not recommended to include such result figure in the discussion. This image can be combined with the distribution points of this study and placed in section 2.1.
21. Lines 361-369: These contents are research results, and the discussion section should not focus on these phenomena, but should explain the reasons for these results. Also, why not propose prevention and control suggestions based on this study? The previous two reviews provided revision suggestions, but the changes were not significant.
If you don't know how to write it, the easiest way is to identify the areas where prevention and control measures should be strengthened. For example, indicate which regions currently have no distribution but the predicted results show high suitability, and which regions currently have no distribution but the predicted results show high suitability in the future.
22. Line 385: This conclusion is very arbitrary, and this study did not investigate these contents.
23. Lines 387-388: It should be placed at the end of this paragraph.
24. Line 389: This sentence is redundant.
25. Line 398: This study cannot draw this conclusion.
26. Line 400-401: These contents seem to overlap with the previous content, and this study cannot draw this conclusion.
Comments on the Quality of English LanguageThere seems to be no special issue with the language, and all the issues with the manuscript are pointed out in the comments.
Author Response
Comments 1: 1. Lines 216-218: The method used in the literature referenced here is the natural-breaks method. If this study also used this method, it should be directly indicated. Although this method is not the best way to determine the threshold for the suitable area, it seems acceptable. However, the classification results of the suitability index in this manuscript should not be exactly the same as the referenced literature, as the results of this study cannot be exactly the same as it. This manuscript needs to calculate the division results of suitability index in ArcGIS based on the research results. |
Response 1: Thank you for pointing this out, it's true that I didn't understand it properly, so I haven't been able to find a proper way to classify it. This time, I readjusted the suitability index based on the characteristics of this species and the classification literature of other species in the same family. (Line 214-217) |
Comments 2: 2. Line 2: The “Cyrotrachelus buqueti (Guer)” should be changed to “Cyrotrachelus buqueti Guer”. |
Response 2: Thank you for pointing out my error. I have corrected the error in the text. (Line 2) |
Comments 3: 3. Lines 33-34: The “Curculionidae” only needs to be repeated once. |
Response 3: Thank you for pointing out my error. I have corrected the error in the text. (Line 31) |
Comments 4: 4. Lines 45-46: This reference ([3]) is still incorrect. The Catalogue of Forest Pests in China needs to be cited here. |
Response 4: Thank you for pointing this out, I have made the change. (Line 43) |
Comments 5: 5. Line 91: The author seems to have misunderstood. The previous round of review pointed out that the third advantage of the MaxEnt model is difficult to understand, rather than the content in lines 85-87. Does the line number pointed out in my comment correspond to the line number understood by the author? |
Response 5: Thank you for pointing this out, I have made the change. (Line 85-87) |
Comments 6: 6. Lines 102-105: Why are these two references cited? These contents seem redundant. The author replied that this part of the content has been deleted, but it has not been deleted. Does the line number pointed out in my comment correspond to the line number understood by the author? |
Response 6: Thank you for pointing this out, I have made the change. (Line 97) |
Comments 7: 7. Line 124: The reference ([5]) cited here does not appear in Table S1. |
Response 7: Thank you for pointing this out, I have made the change. (Line 117) |
Comments 8: 8. Line 139: The SDM tools requires citations, you can refer to http://www.sdmtoolbox.org/technical-info. |
Response 8: Thank you for pointing this out, I have made the change. (Line 134-135) |
Comments 9: 9. Line 164: The MaxEnt software requires citations and the version number should be indicated. If it appears again in the following text, no citation is required. |
Response 9: Thank you for pointing this out, I have made the change. (Line 161) |
Comments 10: 10. Line 165: It should be noted here that when the contribution rate of the variable is lower than what, it will be excluded. |
Response 10: Thank you for pointing this out, I have removed the content. (Line 162-163) |
Comments 11: 11. Line 166: The ENMTools.pl software requires citations. |
Response 11: Thank you for pointing this out, I have made the change. (Line 122) |
Comments 12: 12. Line 172: The number of variables in the title of Table 2 is incorrect. |
Response 12: Thank you for pointing this out, I have made the change. (Line 70-72) |
Comments 13: 13. Line 204: The number of variables is incorrect. |
Response 13: Thank you for pointing this out, I have made the change. (Line 169) |
Comments 14: 14. Lines 241-244: Is it necessary to indicate whether these results are based on the adaptability index of 0.08 or 0.49? Or indicate whether it is based on low habitat suitability or highly habitat suitability? |
Response 14: Thank you for pointing this out, I have made the change. (Line 234-235) |
Comments 15: 15. Line 267 (Figure 4): It is suggested to fully display the South China Sea in the figure on the right. Additionally, the latitude and longitude of the image are blurry. |
Response 15: Thank you for pointing this out, I have made the change. (Line 266) |
Comments 16: 16. Line 289: There is no need for a space at the end of this sentence. |
Response 16: Thank you for pointing this out, I have made the change. (Line 291) |
Comments 17: 17. Line 291 (Figure 5): The latitude and longitude of the image are blurry. |
Response 17: Thank you for pointing this out, I have made the change. (Line 291) |
Comments 18: 18. Line 296 (Figure 6): It is suggested to fully display the South China Sea in the figure. Additionally, the latitude and longitude of the image are blurry. |
Response 18: Thank you for pointing this out, I have made the change. (Line 297) |
Comments 19: 19. Line 303 (Figure 7): The latitude and longitude of the image are blurry, and the points on the right are very small, making it difficult to see what shape each point is. |
Response 19: Thank you for pointing this out, I have made the change. (Line 301) |
Comments 20: 20. Line 359 (Figure 8): The distribution points are blurry, and many of them cannot be seen at all, and the latitude and longitude are also blurry. It is not recommended to include such result figure in the discussion. This image can be combined with the distribution points of this study and placed in section 2.1. |
Response 20: Thank you for pointing this out, I have made the change. Because the distribution points are large and concentrated, some of the distribution points are mixed together. Enlarging a point will cause the points to be connected to each other, and making the point smaller will make it less recognizable in the diagram. (Line 126) |
Comments 21: 21. Lines 361-369: These contents are research results, and the discussion section should not focus on these phenomena, but should explain the reasons for these results. Also, why not propose prevention and control suggestions based on this study? The previous two reviews provided revision suggestions, but the changes were not significant. If you don't know how to write it, the easiest way is to identify the areas where prevention and control measures should be strengthened. For example, indicate which regions currently have no distribution but the predicted results show high suitability, and which regions currently have no distribution but the predicted results show high suitability in the future.
|
Response 21: Thank you for pointing this out, I have made the change. I have removed the reference to the findings of the study and added recommendations for prevention and control based on the study. (Line 358-384) |
Comments 22: 22. Line 385: This conclusion is very arbitrary, and this study did not investigate these contents. |
Response 22: Thank you for pointing this out, I've removed this part. (Line 386) |
Comments 23: 23. Lines 387-388: It should be placed at the end of this paragraph. |
Response 23: Thank you for pointing this out, I've removed this part. (Line 393-397) |
Comments 24: 24. Line 389: This sentence is redundant. |
Response 24: Thank you for pointing this out, I've removed this part. (Line 386-397) |
Comments 25: 25. Line 398: This study cannot draw this conclusion. |
Response 25: Thank you for pointing this out, I've removed this part. (Line 386-397) |
Comments 26: 26. Line 400-401: These contents seem to overlap with the previous content, and this study cannot draw this conclusion. |
Response 26: Thank you for pointing this out, I've removed this part. (Line 386-397) |

Reviewer 2 Report
Comments and Suggestions for Authors
In this revision, the authors have addressed some of the issues raised in previous reviews. However, they still do not specify the geographic extent used for training their model.
I am confused as to why the results look so different between the last revision and this one. How were the methods changed?
Author Response
Comments 1: In this revision, the authors have addressed some of the issues raised in previous reviews. However, they still do not specify the geographic extent used for training their model. I am confused as to why the results look so different between the last revision and this one. How were the methods changed? |
Response 1: Thank you for your comments. This study is based on the prediction of environmental factors on a global scale, because the distribution range of this species is relatively small compared to the global scale, and it only occurs in China and Thailand, so the distribution range of the participating sites in the training model is also concentrated in China and Thailand. Other countries may not be currently invaded by the species, or there may be a lack of research data and therefore no distribution points have been recorded. This set of reasons does lead to a large area of unsuitable habitat, resulting in errors. However, documenting the occurrence of a species is a difficult task in itself, and one of the advantages of the MaxEnt model is that it can accurately predict its potential range using a small sample size. I believe that this study can be used to predict the potential global distribution of the species' range. The two revisions differ because reviewer 1 felt that the methodology used to screen for environmental factors in the previous version of the study was not appropriate for this species. Therefore, we modified the methodology for screening environmental factors by removing only irrelevant environmental factors by removing those that contributed less than 1% and calculating Pearson correlation coefficients. This time, we retained 9 environmental factors, and the distribution range of the habitat of this species changed when the environmental factors involved in the modeling were changed. Therefore the results of these two revisions are partially different. |
